# Large Language Models of Code Fail at Completing Code with Potential Bugs

**Tuan Dinh**[1][*][†]   **Jinman Zhao**[2][*]    **Samson Tan**[2]    **Renato Negrinho**[2]
**Leonard Lausen**[2]    **Sheng Zha**[2]    **George Karypis**[2]
[1]University of Wisconsin–Madison    [2]Amazon Web Services
`tuan.dinh@wisc.edu`
`{jinmaz,samson,renatoni,lausen,zhasheng,gkarypis}@amazon.com`

## Abstract

Large language models of code (Code-LLMs) have recently brought tremendous advances to code completion, a fundamental feature of programming assistance and code intelligence. However, most existing works ignore the possible presence of bugs in the code context for generation, which are inevitable in software development. Therefore, we introduce and study the *buggy-code completion* problem, inspired by the realistic scenario of real-time code suggestion where the code context contains *potential bugs* – anti-patterns that can become bugs in the completed program. To systematically study the task, we introduce two datasets: one with synthetic bugs derived from semantics-altering operator changes (buggy-HumanEval) and one with realistic bugs derived from user submissions to coding problems (buggy-FixEval). We find that the presence of potential bugs significantly degrades the generation performance of the high-performing Code-LLMs. For instance, the passing rates of CODEGEN-2B-MONO on test cases of buggy-HumanEval drop more than 50% given a single potential bug in the context. Finally, we investigate several post-hoc methods for mitigating the adverse effect of potential bugs and find that there remains a significant gap in post-mitigation performance.[3]

## 1   Introduction

Suggesting code for a given context is a frequently used feature in modern integrated development environments (IDEs) [1], bringing productivity gains to the code-writing process. This task is widely studied as code completion [2, 3] in the literature, with techniques and models ranging from probabilistic or sequence modeling [4, 5], incorporating code structure as prior knowledge [6, 7], to adopting deep neural networks [8] and pre-training techniques [9] to learn representations for code. Recently, large Transformer-based language models of code (Code-LLMs) [10, 11, 12] have become a promising paradigm for code completion, attaining state-of-the-art (SotA) performance in various code learning tasks including code completion and generation.

However, existing works studying Code-LLMs often assume the absence of bugs, despite the frequent occurrences and the cost of bugs in software development: on average, 70 bugs are created per 1000 code lines [13]; and fixing bugs costs 50% of development time [14]. Consider a practical coding scenario. A developer wants to use the code suggestion feature in an IDE when writing code. With a high probability, their real-time code context, as an input to the code predictor, contains typos or less refined, potentially buggy implementations. Since bugginess is a property of a complete program,

---

[*]Equal contribution.

[†]Work done while interning at Amazon Web Services.

[3]Code and datasets are available at `https://github.com/amazon-science/buggy-code-completion`

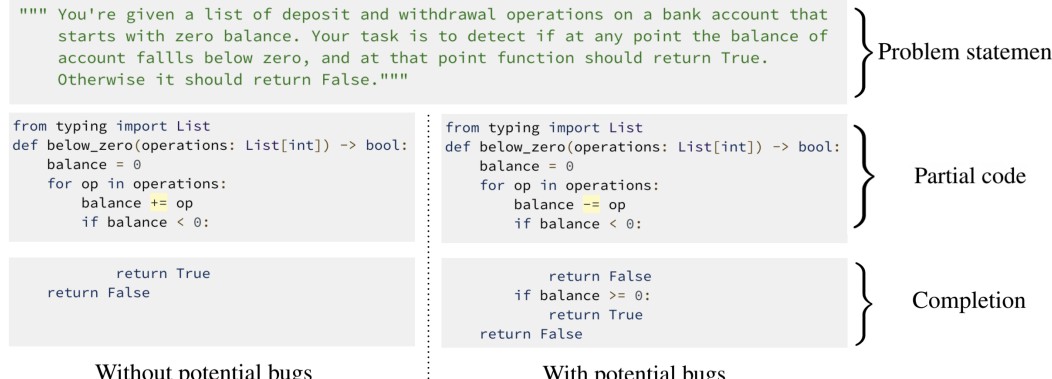

Figure 1: **Illustrations for code completion (left) and buggy-code completion (right).** (top) problem statement for function implementation, (middle) partial code with (right) or without (left) potential bugs (highlighted), (bottom) code completions from CODEGEN-2B-MONO [12]. The completed code is functionally correct on the left but incorrect on the right, failing test case `below_zero([1, 2]) == False`. The example is based on `HumanEval/3` from buggy-HumanEval.

it is not well-defined how to detect bugs and repair the buggy code in this short and incomplete code context, making the application of existing bug-detection or code-repair tools sub-optimal or infeasible. It is also worth mentioning that the gap between the *in vitro* and *in vivo* performances of code completion models [15, 16, 17] remains large. Therefore, a natural question arises: *Can existing Code-LLMs provide good code suggestions given the unrefined nature of draft code?*

To answer this question, we introduce and study the problem of completing code with potential bugs in the code context, dubbed *buggy-code completion (bCC)*. In this study, we focus on using Code-LLMs to generate functional implementations from a code context, where the code context consists of a problem specification and a piece of partial code.[4] A *potential bug* in a piece of partial code is a code span that can become a bug provided some completion, *i.e.,* it fails the completion and at the same time can be changed to make the completion work. Note that a potential bug is not a bug per se without a completion. Shown in Figure 1 is an illustration of our task with a problem description at the top and partial code in the middle. On the left are a reference code context and a correct completion from the chosen Code-LLM. On the right, the highlighted potential bug (-=) makes the reference completion incorrect. The Code-LLM reacts to this potential bug by generating a different completion (bottom right). However, the completed code is still functionally incorrect.

To conduct a quantitative study of bCC, we construct two datasets. First, *buggy-HumanEval* dataset contains interview-style coding problems from HumanEval dataset [10], with buggy/reference partial code pairs being generated by introducing semantic-altering operator changes to reference solutions. This dataset provides a well-controlled setting to assess models' behavior upon potential bugs. Second, *buggy-FixEval* dataset, based on FixEval [19], contains user submissions to coding-contest problems. The buggy/reference pairs are constructed from rejected and accepted submissions by the same user to a given problem. This dataset helps assess models' performance over a realistic distribution of potential bugs. Our benchmarks are well associated with the existing benchmarks for Code-LLMs.

Via our empirical studies, we find that the presence of potential bugs drastically degrades the code-completion performance of high-performing Code-LLMs, with test-case pass rates dropping to below 5% across both datasets for all tested model variants. For instance, on buggy-HumanEval, the test-case pass rate of CODEGEN-2B-MONO completions drops from $54.9\%$ (reference partial code) to $3.1\%$ (partial code contains potential bugs), which is worse than the score when no partial code is provided ($9.3\%$). Our results demonstrate that Code-LLMs are highly susceptible to potential bugs.

Furthermore, we attempt several post-hoc methods to augment Code-LLMs to better deal with potential bugs, namely removal-then-completion, completion-then-rewriting, and rewriting-then-completion. The latter two augment the Code-LLMs with an external code repairer as the rewriter component. Our evaluation shows that the attempted methods improve the buggy-code completion performance of all tested Code-LLMs. However, the performance gap remains large between these

---

[4]This setting is well-aligned with the text-to-code generation task for Code-LLMs [12, 18].

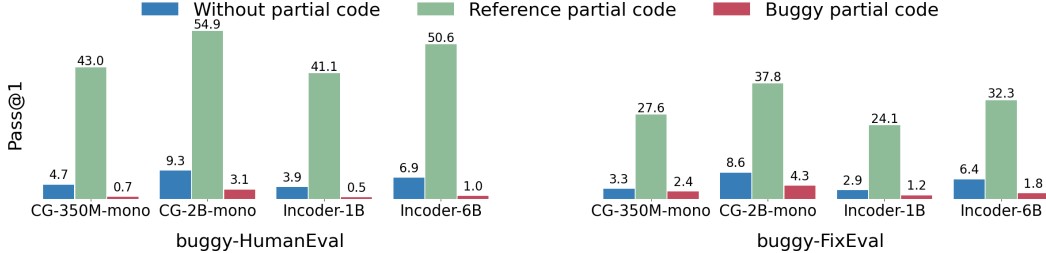

Figure 2: **Performance (`pass@1` (↑)) degradation in the presence of potential bugs**. CG: CODE-GEN. Potential bugs severely harm the completion of all CodeLMs (red bars) compared to non-buggy settings (teal bars), making it even worse than completion without partial code (blue bars).

methods and the completion with reference partial code. We provide further case studies and analyses, *e.g.,* effects of potential bugs' locations or the successful cases of naïve completion for a better understanding of the behavior of the tested models in buggy-code completion.

**Study scope and contributions.** This work aims to explore and understand the behaviors of Code-LLMs under the buggy-code completion setting. We (i) define the novel buggy-code completion task, (ii) introduce two representative benchmark datasets, (iii) demonstrate the inability of Code-LLMs to handle potential bugs, and (iv) evaluate several baseline methods for improving Code-LLMs on buggy-code completion.

## 2 Buggy-Code Completion

In this work, we consider the code completion setting with inputs consisting of (1) a specification $h$ that specifies the desired functionality of the program and (2) a code context $s$, which is some unfinished code to be completed. Here, $h$ is given as a docstring or a problem statement in English, and $s$ is given as a few lines of code that are the beginning part of a program, which we refer to as *partial code* or *(code) prefix*. In the conventional setting of code completion, the objective is to suggest a completion $c$ such that $t := s :: c$ is a program that satisfies $h$, where "::" denotes code concatenation. Our *buggy*-code completion setting extends the conventional one with a challenging and realistic consideration: $s$ may contain *potential bugs*.

Consider a real-world scenario where a programmer works on unfinished code and utilizes the code auto-completion feature for suggestions. Note that the coding process often sees coding mistakes or introduces inconsistent code usages, which are not necessarily "incorrect" per se. However, there is a fair chance that such mistakes or inconsistencies cause unintended and undesirable behaviors in the final program. In other words, they become bugs. Based on this intuition, we define potential bugs as the following:

**Definition 2.1** (*potential bug*). Consider a specification $h$ and a reference code prefix $s$ for which some completion $c$ exists such that $t := s :: c$ satisfies $h$. A *potential bug* is manifested as a small edit $e$ over $s$ such that $t' := s' :: c$ does not satisfy $h$, where $s'$ is the result of applying $e$ on $s$.

We note that potential bugs are not bugs per se and are only defined with respect to some reference code prefix $s$ and completion $c$. The code prefix $s'$ containing potential bugs is referred to as "potentially buggy" or simply "buggy" prefix (with respect to $s$) throughout the paper. In most cases, we omit the "with respect to" part for brevity. The resulting term "buggy prefix" refers to the definition here and by no means assumes that the prefix itself is buggy. Our study focuses on potential bugs associated with semantic bugs, *i.e.,* edits that do not introduce syntax errors, as semantic bugs are generally more challenging and interesting than syntax ones.

*Remark* 2.2. Intuitively, we assume that the reference prefix $s$ highly correlates to preferred (practical) implementations of the task and that any deviation from $s$ is likely less preferred. We use the operational definition of potential bugs in 2.1 for its simplicity and verifiability, as well as for that it allows the initial exploration of the previously under-investigated scenario of bCC. We note that by this definition, a reference prefix $s$ itself can, in some cases, albeit less likely, be "buggy" with respect to some other references. However, such a case is less of a concern in this study, as we are mostly interested in finding a completed functional program out of $s$ or $s'$.

**Definition 2.3** (*buggy-code completion, bCC*). Given a specification $h$ and a code prefix $s'$ containing potential bugs, *buggy-code completion* is the task of generating a complete program $t$ that satisfies $h$.

We deliberately loosen the constraint that $t$ should contain $s'$ as a prefix, as by definition, and as well as one can see from the example in Figure 1, fixating on the buggy prefix can make suggesting a satisfying solution difficult, if not impossible. As we explore in Section 4.3, allowing models to suggest fixes to the buggy code prefix significantly increases the possibility that a generated program passes the tests. However, it is often still possible to continue a buggy prefix to a valid program solution (see an example in Figure 12).

An alternative view of the setting is that the buggy code prefix provides a noisy and potentially flawed precondition for generating a solution to the coding problem. It presents a good-faith effort from a user to instruct the tool about their intended code solutions. A good-behaving model should take this as a hint and, at worst, discard it so that the generation performance is not worse than when no code prefix is given. However, this is not the case for our evaluated models (Section 4.2).

*Remark* 2.4. Our bCC formulation is *not* a simple combination of code repair and code completion. As bugginess is a property of completed programs, repairing the partial code is an ill-defined problem, thus making repairing-then-completion also ill-defined. Therefore, our bCC task is better viewed as an extension of code completion with the additional challenge that generating semantically correct continuations from the given partial code without deviation may be difficult or even infeasible. This challenge requires models to be aware of the existence of potential bugs for better suggestions.

# 3 Benchmarks

This section introduces our new datasets with proposed baseline methods and evaluation metrics.

## 3.1 Datasets for bCC

Based on our task definition, each instance in a bCC benchmark should contain a problem description specifying requirements, a piece of buggy code prefix to be completed, and a set of test cases for assessing the correctness of the finished code. Optionally, an instance also has a corresponding reference code prefix and a valid solution completed from the reference code. To our knowledge, no existing dataset meets all the desiderata. Commonly used large-scale datasets for code completion, *e.g.*, Py150 [20] and Java Corpus [21] do not come with test cases. Recent small-scale manually curated datasets for code generation, *e.g.*, HumanEval [10], MBPP [18], APPs [22], come with test cases and reference solutions but no buggy or failed solutions. There is also no predefined way to extract partial code from the reference solutions. This is also the case for program-repair datasets: popular datasets either lack test cases, *e.g.*, [23, 24, 25, 26] and/or are of small sizes, *e.g.*, [27, 28, 29].

We introduce two datasets for evaluating bCC in Python, both with all the aforementioned desired components. We ensure that 1) all partial code snippets contain potential bugs that fulfill our definition, *i.e.*, their respective completed code is incorrect, certified by failing test cases; and 2) all potential bugs are semantic in nature, *i.e.*, they cause no syntax error in their respective completed code.

### 3.1.1 Buggy-HumanEval

We first introduce buggy-HumanEval to offer a controlled setting for evaluating bCC in the presence of a single semantic bug. Buggy-HumanEval contains 1896 bCC instances constructed from a subset of HumanEval problems [10]. The HumanEval dataset is a popular dataset of manually written introductory coding problems designed for evaluating the code generation ability of Code-LLMs.

Potential bugs are introduced as semantic-altering operator changes to reference solutions. We search for applicable binary operators in reference solutions and change them into their semantic opposites, *e.g.*, + into −. To ensure that the edit introduces a bug, we execute the altered program and only keep the ones failing some tests. We then specify a line after the altered operator to split the solution into a code prefix and suffix. We keep the buggy prefix as part of the input for bCC. We split the unaltered solution at the same line to get a corresponding reference prefix. On average, the problems selected for buggy-HumanEval have longer solutions than the unselected ones in HumanEval, with 8.2 lines vs. 2.9 lines of code, respectively. Appendix A.1 provides details of our dataset.

### 3.1.2 Buggy-FixEval

To evaluate bCC with more realistic bugs, we introduce buggy-FixEval with bCC instances constructed from CodeNet [30] and FixEval [19]. FixEval is a program repair benchmark based on user submissions to competitive programming websites. Each data sample in FixEval can be viewed as a pair of submitted programs from the same user, with the accepted submission being regarded as reference or fixed and the preceding rejected submission being regarded as buggy.

To create buggy-FixEval, we match and pair each problem with its problem statement found in CodeNet and omit the problems with no match. For each submission pair, we identified potential bugs as the differences between the rejected and accepted submissions. We ensure that the rejected submission contains no syntax error and fails at least one test case. We split the solutions into halves and regard the prefix from the rejected solution as containing potential bugs and the prefix from the accepted solution as a reference. To guarantee the differences between the buggy prefix and the reference prefix related to potential bugs, we impose a limit on the character-level edit distance between them, ignoring comments and white spaces. The lower limit ensures that the differences are not comments or white spaces. The upper limit (20, chosen by manual inspection) reduces the chance that the difference is related to a reformatting or a re-implementation. We then manually inspected all the remaining pairs and excluded undesirable cases such as that the reference prefix itself is already a correct solution or that the two prefixes are semantically equivalent. Finally, we execute the concatenation of the buggy prefix and the reference completion and ensure that it fails at least one test case. More details about creating buggy-FixEval can be found in Appendix A.2.

### 3.2 Baseline Methods for bCC

Beyond the naïve generation from code context, we study several methods for augmenting Code-LLMs to utilize the partial code better while mitigating the effect of potential bugs. Assuming a bug detection component is equipped, we focus on simple and modular-design approaches that do not require additional training and are flexible for further incorporating newly developed Code-LLMs.

**Removing partial code, then completing (removal-then-completion).** We bypass the negative effect of buggy partial code by removing the entire content code fragment from the model input. In particular, for buggy-HumanEval, the input to Code-LLMs after removal consists of the problem statement and the function header (and examples). For buggy-FixEval, we keep only the statement and the code fragment used for reading input data. Intuitively, as removal-then-completion guarantees that the input to Code-LLMs contains no bug, we expect this method to perform better than the naïve completion. However, the drawback of removal-then-completion is its sacrifice of all potentially useful information brought by the partial code.

**Completing first, then rewriting the program (completion-then-rewriting).** This approach attempts to fix the buggy code using pre-trained code-repair models [31, 32, 25], *e.g.,* neural translating a buggy program into a valid program [31]. As code-repair models, or code fixers, are usually trained to fix the complete programs, we first complete the code by naïve completion, then apply code fixers on the potentially buggy programs. We use RealiT [32], a SotA program repair model as the code fixer. RealiT is designed for misused variables, wrong literal, wrong binary, and unary operators, which is especially suitable for the bugs introduced in the buggy-HumanEval.

**Rewriting the partial code, then completing (rewriting-then-completion).** This approach attempts to resolve potential bugs in the partial code before completing. To do so, we first locate code lines containing the potential bug, then rewrite these lines. To detect potential bugs, we propose a likelihood-based measure to identify the line most likely to contain a potential bug. In particular, we score each code line with the following procedure. First, for each token, we define its buggy score to be the difference in likelihoods between the token with the highest likelihood (*i.e.*, the `argmax` token) and the observed token. The buggy score for each line is then calculated by taking either the maximum or average of non-zero buggy scores of its tokens. The line with the highest score is most likely to contain the potential bugs. We use the INCODER-6B [33] as the infilling language model for rewriting code. We provide a detailed explanation and example illustration of likelihood-based measures in Appendix B to understand our rewriting-then-completion method better.

Table 1: `Pass@1` (↑) of completion methods on buggy-HumanEval and buggy-FixEval datasets. For all Code-LLMs, all three proposed methods improve the completion performance of the naive completion. On average, completion-then-rewriting and rewriting-then-completion achieve the best scores on buggy-HumanEval and buggy-FixEval, respectively. Nevertheless, there are still substantial gaps between these methods and completion with reference prefixes. *Best methods in each column are in **bold**. Table 7 in Appendix C provides results with a larger model (*CODEGEN-16B-MONO*).

| Prefix | Method | buggy-HumanEval | | | | buggy-FixEval | | | |
| --- | --- | --- | --- | --- | --- | --- | --- | --- | --- |
| | | CODEGEN- | | INCODER- | | CODEGEN- | | INCODER- | |
| | | 350M | 2B | 1B | 6B | 350M | 2B | 1B | 6B |
| reference | completion | 43.0 | 54.9 | 41.1 | 50.6 | 27.6 | 37.8 | 24.1 | 32.3 |
| buggy | completion | 0.7 | 3.1 | 0.5 | 1.0 | 2.4 | 4.3 | 1.2 | 1.8 |
| | removal-then-completion | 4.7 | 9.3 | 3.9 | 6.9 | **3.3** | **8.6** | **2.9** | **6.4** |
| | rewriting-then-completion | 14.1 | **24.9** | 9.1 | 16.4 | 2.4 | 7.2 | 2.6 | 5.1 |
| | completion-then-rewriting | **22.5** | 23.6 | **22.7** | **25.2** | 2.3 | 4.7 | 1.7 | 3.0 |

## 3.3 Evaluation Metrics

We measure the functionality of a completed program by executing it against the provided test cases. Following the recent code-generation works [*e.g.*, 10, 34, 12, 33], we measure the `pass@k` (↑) for each bCC instance as $\text{pass@}k := 1 - \binom{n-c}{k}/\binom{n}{k}$ where $n$ completions are sampled from a model, and $c$ of them pass all the tests. We choose $n = 100$ and $k = 1, 10, 100$. This metric estimates the probability that any of $k$ samples from the model passes the tests. As multiple bCC instances may be derived from the same programming problem, we first average the `pass@`$k$ within each problem, then average across all problems (macro-average) to avoid the dominance of a few problems.

While passing all test cases does not 100% guarantee the correctness of a program, this metric provides a practical and efficient proxy for functional correctness. Note that we do not use match-based metrics, *e.g.*, exact match or CodeBLEU [35] because they do not properly reflect the functionality of generated code [22, 10, 18], and no reference completion is available for buggy prefixes.

## 4 Experiments

We design two experiment sets to investigate (1) how well the existing Code-LLMs adapt to bCC (Sec. 4.2) and (2) whether we can have a simple fix with Code-LLMs for bCC (Sec. 4.3). We provide ablation studies on potential bugs (Sec. 4.5) and on combining buggy-based completion with reference prefix (Sec.C.4). Sec. 4.6 presents case studies about interesting behaviors of Code-LLMs under bCC, with additional results in Appendix C.

### 4.1 Experiment Settings

**Code-LLMs.** We evaluate the two popular and open-sourced Code-LLMs for code completion. **CODEGEN** [12] is a family of LLMs trained on both natural and programming language corpora with high performance in code generation on HumanEval [10]. We use released model checkpoints: CODEGEN-350M-MONO, CODEGEN-2B-MONO, and CODEGEN-16B-MONO. **INCODER** [33] models are trained with a causal masking objective, allowing them to fill blocks of code conditioned on arbitrary left and right contexts. We use the released model checkpoints INCODER-1B and INCODER-6B, each with 1B and 6B parameters. We select CODEGEN [12] and INCODER [33] since they are publicly available with high performance of code generation.

**Generating completions.** *Input format.* For buggy-HumanEval, a model is expected to complete an unfinished function. Following the HumanEval benchmark [10], we set the models' input as the partial code leading up to the completion location, with the problem description embedded as the docstring of the unfinished function. For buggy-FixEval, the completion task is to complete an unfinished program with inputs being the problem description as a file-level docstring (quoted within triple quotation marks), followed by the partial code leading up to the completion location. *Generation and sampling details.* Following the best-performing settings reported in the corresponding works [12, 33], we use temperature sampling with temperature = 0.6 for CODEGEN and top-$p$ sampling with $p = 0.95$ and

temperature = 0.2 for INCODER. Based on the reference solutions and computing efficiency, we set the maximum length limit for outputs from 200 to 600 tokens, varying with the problem sets. We observed similar performance trends between the tested models across different length settings. We post-process the output string following the same procedure used in their code releases.

**Code-repair and code-rewriting models.** For removal-then-completion, we use the latest model of RealiT [32] as the code-fixer, which is trained and tested over artificial and realistic bugs. For the code-rewriting model, we use the INCODER-6B model [33] due to its ability to code infilling and apply the similar settings used for infilling reported in the model's paper [33].

## 4.2 How Well Do Existing Code-LLMs Perform on Buggy-Code Context?

We evaluate the latest Code-LLMs for buggy-code completion on buggy-HumanEval and buggy-FixEval, shown in Figure 2. In particular, the performance is measured in terms of `pass@1`, and we use four models: CODEGEN-350M-MONO, CODEGEN-2B-MONO, INCODER-1B, and INCODER-6B. First, comparing the scores between reference and buggy partial code for each model, we see that the presence of potential bugs is detrimental to the completion models, with `pass@1` drops from 41.1–54.9% to 0.5–3.1% over buggy-HumanEval, and from 24.1–37.8% to 1.2–4.3% over buggy-FixEval. Moreover, the `pass@1`'s upon buggy partial code is universally dominated by those without partial code: 3.9–9.3% over buggy-HumanEval, 2.9–8.6% over buggy-FixEval. Table 7 (Appendix C) shows the similar findings with a very large model (CODEGEN-16B-MONO). These results indicate that (i) the tested Code-LLMs drastically fail at bCC instantiated by our datasets, and (ii) the presence of potential bugs destroys the benefit brought by the partial code.

**Why do Code-LLMs fail at bCC?** We manually inspect samples from the best-performing CODEGEN-2B-MONO model and find the two most common failure modes among failed samples. First, the model fails to react to the potential bugs (*i.e.,* common completions remain the same), as shown in Figure 8. This mode happens in 90% of instances and 93% of problems with at least one failed instance. We conjecture that the model is not sensitive to and thus ignores minor code changes and/or chooses to default to common patterns in the training data. Secondly, the model fails to bypass the potential bugs, likely because such patterns are rare in high-quality code. In other words, the model might have recognized the potential bugs and significantly changed the output distribution but still failed. Figure 7 illustrates an example of this case. We provide further details in Appendix D.2.

## 4.3 How Effective Are Baseline Completion Methods Against Potential Bugs?

We evaluate the completion methods introduced in Section 3.2 using the same bCC setting, shown in Table 1. We include results of completing reference partial code to see how potential bugs affect the Code-LLMs. Figure 6 in Appendix C provides full results and interpretations for $k = 1, 10, 100$.

**The effect of proposed completion methods.** All three proposed methods outperform the naïve buggy-code completion baseline. On buggy-HumanEval, we observe the general trend of completion-then-rewriting outperforming rewriting-then-completion, which outperforms removal-then-completion. Note that these scores of removal-then-completion are generally lower than reported scores of the similar method [12, 33] probably because buggy-HumanEval is derived from a relatively more challenging subset of HumanEval (see Section 3.1.1). On buggy-FixEval, we observe removal-then-completion to outperform rewriting-then-completion, which outperforms completion-then-rewriting. The performance gap increases as the size of the completion model rises. Similar comparison trends are observed for `pass@k` for $k = 10, 100$. Nevertheless, performance gaps remain significant between the best method and the completion from the reference code for all settings.

**The effect of Code-LLMs' capacity.** For each type of code completion model, the larger version performs better than the smaller version using the same method. For instance, CODEGEN-2B-MONO outperforms CODEGEN-350M-MONO for all settings in two datasets. Compared to CODEGEN, INCODER models, in general, obtain better `pass@1` but worse `pass@10` and `pass@100`. This suggests that CODEGEN models generate more diverse completions, while INCODER models generate more precise completions. Furthermore, INCODER is more sensitive to buggy partial code than CODEGEN models, evidenced by the lower scores from naïve bCC.

Table 2: Comparing `pass@1` (↑) of completion methods on reference and buggy partial code. While the proposed mitigation methods achieve better performances than naïve completion with buggy prefixes, they may harm the completion when no bug exists (reference).

| Dataset | **buggy-HumanEval** | | | | **buggy-FixEval** | | | |
|---|---|---|---|---|---|---|---|---|
| CODEGEN- | 2B-MONO | | 350M-MONO | | 2B-MONO | | 350M-MONO | |
| | reference | buggy | reference | buggy | reference | buggy | reference | buggy |
| naïve completion | 54.9 | 3.1 | 43 | 0.7 | 37.8 | 4.3 | 27.6 | 2.4 |
| rewriting-then-completion | 49.6 | 24.9 | 35.9 | 14.1 | 37.0 | 7.2 | 26.4 | 2.4 |
| completion-then-rewriting | 27.7 | 23.6 | 22.2 | 22.5 | 19.4 | 4.7 | 3.7 | 2.3 |

Table 3: Balancing rewriting-then-completion for the buggy and reference settings with bug-detection thresholding. Completion performances can be adjusted via varying bug-detection thresholds from 0 (fix all partial codes) to 1 (keep all partial codes). Results are with CODEGEN-2B-MONO.

| **buggy-FixEval** | threshold | | | |
|---|---|---|---|---|
| partial-code setting | 0 | 0.3 | 0.9 | 1 |
| reference | 8.6 | 14.6 | 37.0 | 37.8 |
| buggy | 8.6 | 7.1 | 7.2 | 4.3 |

**Synthetic bugs versus real bugs.** Among the two bCC datasets, we observe that the overall performance of mitigation methods is better on buggy-HumanEval than buggy-FixEval. This indicates the difficulty of realistic potential bugs in buggy-FixEval: There may be multiple bugs; bugs may be potentially mixed with non-bug-fixing changes; and bugs are more nuanced than single operator changes. Furthermore, while achieving the best performance in most cases, completion-then-rewriting only shows marginal differences from other methods when using larger models on buggy-FixEval.

**Take-away:** *Our baseline methods improve the completion for all evaluated Code-LLMs. However, the remaining performance gaps to the completion with reference partial code are still large.*

### 4.4 What If Partial Code Does Not Have Potential Bugs?

As shown in Table 2, mitigation methods for bCC may harm completion from reference code context (removal-then-completion is the same for both settings, thus not listed.) This suggests that a general code completion should consider both cases when potential bugs may and may not exist.

With our baselines, we can use the thresholding approach for detecting potential bugs. For instance, in rewriting-then-completion, a token is only considered a potential bug if its likelihood gap to the `argmax` token is beyond a threshold (between 0 and 1). Table 3 compares `pass@1` scores of rewriting-then-completion varying thresholds on buggy-FixEval. We can see that the threshold of 0.9 can help achieve a relatively good balance for the two cases. A similar approach can be applied to completion-then-rewriting, as RealiT provides the probability of a token being a bug.

*Remark* 4.1 (*Imbalance in real distributions*). We note that distributions of natural code bugs are usually imbalanced as practical bugs occur infrequently [36, 37]. However, potential bugs may occur more frequently as our target setting is a work-in-progress code scenario [38] rather than high-quality code of popular open-source projects in previous studies. Nevertheless, for comparing methods, we evaluate more balanced data to avoid the dominant effect of performance on reference code.

### 4.5 Analysis of Effect from Bug and Split Locations

To understand how bug location and context size affect completion, we aggregate results by the location of potential bugs and the location of partial code splits within buggy-HumanEval. The locations are normalized as (potential bug line #)/(# lines) and (split line #)/(# lines), where potential bug line #, split line #, and # lines are the number of lines starting from the function header to the line containing the potential bug, to the end of the partial code, and to the end of the canonical solution.

Figure 3 presents heatmaps of `pass@1` scores (averaged over the bCC instances falling into each cell) evaluated on the CODEGEN-2B-MONO with naïve completion on reference HumanEval, and naïve

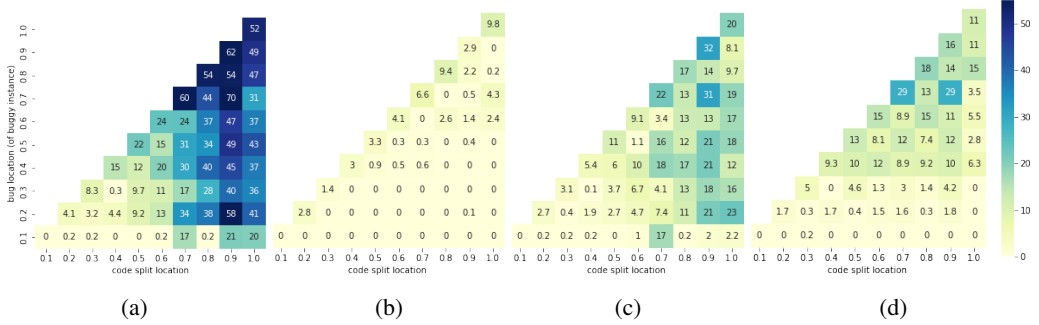

Figure 3: **Average `pass@1` scores by bug and code split locations using CODEGEN-2B-MONO on buggy-HumanEval**. *Left to right*: (a) naïve completion on reference code, (b), (c), (d) naïve completion, completion-then-rewriting, and rewriting-then-completion on buggy code. Locations are normalized by the length of reference solutions. For each split location in (a), the scores may vary across potential bug locations as the buggy-instance distribution is non-uniform across bug locations.

completion, completion-then-rewriting, and rewriting-then-completion (max) on buggy-HumanEval. First, the model performs better given longer partial code in the reference case (Figure 3a). While the naïve completion performs overall poorly with potential bugs (Figure 3b), it performs relatively well when potential bugs appear on or near the last line of the partial code (along the diagonal). More interestingly, rewriting-then-completion achieves higher scores when potential bugs appear later (Figure 3d), while completion-then-rewriting performs better with longer code prefixes (Figure 3c). We suspect that a longer prefix makes completion models less likely to deviate from the reference completion. Hence, as inputs to the subsequent rewriting model, the completed code better resembles input types for which the rewriting model is trained, making the repairing more likely to succeed.

### 4.6 Case Studies

**Are potential bugs always harmful?**  As discussed in Section 2, potential bugs do not guarantee the completed code to be buggy. While we observe performance degradation of Code-LLMs under the presence of potential bugs, we find several interesting cases where the models manage to generate correct code. For instance, Figure 12 in Appendix shows that for the potential bug == (highlighted) modified from `!=`, the completion model deviates its original algorithmic flow with `continue` command and completes with correct functionality, albeit different from the canonical solution. This is an example that some bCC cases are recoverable and that Code-LLMs can adapt to them.

**When do Code-LLMs succeed at bCC?**  For the successful cases of naïve completion at bCC, we observe that either the model (i) ignores the incorrect state and generates the correct completion or (ii) takes into account the potential bug to generate a completion adapting to it. Figure 9 in Appendix shows an example when Code-LLMs ignore the `if-else` statement to bypass the potential bug. Further case studies and more elaborate discussions are in Appendix D.

## 5 Related Work

**Code completion.**  Code completion provides code suggestions based on a given context [39, 40, 38]. The scope of completion ranges from the next token or line [11], method and class names [41], to the entire of the function [42] or program. Early works [5, 43, 4] viewed code as sequences of tokens and applied statistical language models to the problem, along with other attempts at building probabilistic models of code [44, 41, 45]. Later works adopted deep neural networks [8, 46] and pre-training techniques [9, 47] for better code modeling and completion. Our work considers the code completion setting at the function and program levels and focuses on using large language models for completion. Beyond sequence modeling, recent works considered integrating code prior knowledge via abstract syntax trees [6, 8, 48], code token types [9], graph structures [7], hierarchical context [49], or generating the sketch [50], or even extending the task's information beyond the given input files [51, 52]. We focus on using only the task description and partial code as input prompts to the model, allowing the use of more Code-LLMs and publicly available datasets.

**Automatic program repair.**   The research on automatic program [53, 54, 55, 56, 57, 58, 59, 25] relieves developers from the enormous effort of finding and fixing programming bugs. Recently, Code-LLMs have been adapted for program repair by translating buggy programs into their reference counterparts [60, 31, 61]. Among those, we use RealiT [32] as our repair model in the completion-then-rewriting method since they obtain the SotA results and utilize similar simple mutations during training. Despite the similarity, code repair commonly targets fixing bugs from *complete* programs while we study potential bugs from partial code. To enrich the amount of data for program repair, methods have been proposed to synthesize artificial bugs through code mutants [62, 63, 64, 65, 25] or learning to create bugs [66]. Similarly, we employ code mutants to create artificial bugs.

**Relation to adversarial examples.**   Adversarial examples are instances where small perturbations to the input lead the model to change into wrong predictions. They have been extensively studied in computer vision [67, 68] and natural language processing [69]. Recent works suggested similar situations for code-learning models, where small, semantic-*preserving* code transformations led to performance degradation [70, 71]. Buggy-code completion can be seen as a dual problem to adversarial examples, where we expect the model to adjust its predictions up on small semantic-*altering* changes in the input. In our case, sensitivity is not a problem, but *in*sensitivity is.

**Benchmarks for buggy-code completion.**   Multiple benchmarks have been studied for code completion and program repair. For *code completion*, CodeXGLUE [11], CodeNet [30], and HumanEval [10] are widely used. CodeXGLUE contains corpora of Java and Python programs for completion but only supports match-based evaluation. CodeNet collects programming problems from online judge sites, with both solutions and test cases. HumanEval can be considered a Python-function completion dataset, with the context being the problem statement and function header. We derive our datasets from HumanEval and CodeNet datasets. For *neural program repair*, many datasets require the match-based evaluation [24, 26] or focus on the compiler errors [23, 25], which are different from our setting. While IntroClass [27], QuixBugs [28], Defects4J [72], or Refactory [29] provide the test suites for evaluation, their test does not reflect the real-world bugs or lacks context support for the use with Code-LLMs [19]. FixEval [19] is recently proposed as a new context-aware program repair dataset to mitigate these limitations, with many problems derived from the real submitted programs. However, as FixEval does not provide the problem statement and focuses solely on the program repair, we derive a new benchmark using FixEval and its source of problems – CodeNet [30].

# 6   Discussion and Conclusion

**Limitations.**   Our baseline methods developed for bCC may degrade the completion performance on reference code context, as shown in Section 4.4, suggesting the need for balancing the buggy and reference settings in solutions to bCC. Furthermore, while buggy-FixEval is associated with real-world coding contest programs, it is unclear how closely buggy-FixEval aligns to the general software development setting where obtaining a test suite and proper evaluation are more challenging.

**Impact and applications.**   As our work focuses on the less refined and more error-prone work-in-progress code, the code context should be viewed as a hint of user intent rather than a high-quality "gold" implementation. It thus naturally follows that a pair programmer or a smart tool should suggest a change to the draft code rather than blindly continue it if they believe a certain part of the existing draft is not intended. From a user experience perspective, an IDE can display code change suggestions to a user's existing code if such parts are identified. Similar functionality already exists for other types of code change suggestions, *e.g.,* spelling correction and missing imports.

**Conclusion.**   We introduce and define the buggy-code completion problem, inspired by the practical coding scenario where one completes a coding program given the problem statement and a partial code with potential bugs. We construct two new datasets, buggy-HumanEval and buggy-FixEval, as task benchmarks and find that the presence of potential bugs significantly degrades the completion performance of all evaluated large language models of code. Our further investigation of completion methods for Code-LLMs in dealing with potential bugs shows that completing with potential bugs remains challenging despite augmenting models with external program-repair models. We provide extensive ablation and case studies for further understanding and analysis of buggy-code completion setting. We hope our novel and systematic study paves the way for future works in understanding and improving the usability of Code-LLMs under practical software-development settings.

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

# A  Dataset Details

This section presents the details of dataset construction and specifications. The length statistics are reported in Tab 5.

## A.1  Buggy-HumanEval

HumanEval[5] [10] is a dataset designed for evaluating code generations from natural-language descriptions. It contains 164 manually written introductory coding problems in Python 3. Each problem is given in the format of a partial program, referred to as a "prompt": a function header with a docstring, sometimes with necessary imports and helper functions before it. The problem description and often a few input-output examples are encapsulated in the docstring. Each problem is accompanied by a separate set of test cases and a manually written function body referred to as the "canonical solution", such that the concatenation of the prompt and the canonical solution passes all the tests. See an example in Figure 5.

To create buggy-HumanEval, we introduce artificial bugs by flipping binary operators in the canonical solutions into their semantic opposites. Specifically, we use the Python `ast`[6] library and consider all operators under `ast.BinOp`, `ast.Compare` and `ast.AugAssign`, e.g. `/`, `>=` and `+=`. For each found operator, we change it to its semantic opposite, *e.g.,* `/` to `*`, `>=` to `<`, or `+=` to `-=`. Table 4 shows a complete list of the operators we considered and their opposites. We then check if the solution fails any test cases after the change. We skip cases where the semantic opposite is ambiguous (*e.g.,* mod `%` vs. floor division `//` or multiplication `*`) or where the altered solution still passes all the tests[7].

Suppose the canonical solution has $L$ lines, and the operator change happens in the $l$-th line. For each $l \leq i < L$, we append the first $i$ lines of the altered canonical solution to the original HumanEval prompt to form a piece of potentially buggy partial code, aka the "buggy" prompt for buggy-code completion. Accompanying each buggy prompt, we also provide a "reference" prompt by concatenating the original prompt with the first $i$ lines of the original canonical solution. In the example in Figure 5, $L = 8$, $l = 3$, $i = 4$. The `!=` in the third line of the canonical solution is changed to `==`.

We generated 1896 buggy-code completion instances from 107 unique HumanEval problems. This procedure results in the list of the number of instances per problem as follows,

[9, 31, 0, 5, 1, 0, 13, 0, 3, 0, 0, 4, 1, 0, 1, 0, 0, 8, 0, 16, 0, 0, 0, 1, 28, 0, 0, 0, 0, 10, 63, 0, 0, 2, 14, 2, 0, 38, 6, 0, 0, 7, 5, 0, 14, 3, 6, 1, 0, 0, 2, 0, 0, 6, 16, 4, 2, 27, 0, 16, 0, 12, 5, 3, 1, 0, 6, 21, 0, 51, 26, 7, 14, 13, 12, 0, 1, 0, 22, 180, 12, 1, 0, 0, 0, 1, 0, 9, 0, 0, 18, 2, 32, 57, 5, 0, 1, 47, 0, 5, 12, 8, 2, 0, 39, 23, 8, 16, 22, 18, 0, 10, 16, 0, 0, 5, 15, 34, 4, 0, 0, 37, 52, 0, 17, 31, 1, 133, 29, 14, 21, 1, 0, 11, 3, 1, 0, 6, 174, 20, 14, 15, 13, 8, 6, 35, 12, 2, 11, 0, 0, 18, 16, 8, 6, 0, 0, 7, 2, 11, 0, 0]

Table 4:  We considered Python 3 operators when introducing bugs to HumanEval.

| ast class | BinOp | | | | AugAssign | | | | Compare | | | | | |
|---|---|---|---|---|---|---|---|---|---|---|---|---|---|---|
| ast token | Add | Sub | Mult | Div | Add | Sub | Mult | Div | Eq | NotEq | Lt | LtE | Gt | GtE |
| Operator | + | – | * | / | += | -= | *= | /= | == | != | < | <= | > | >= |
| Opposite | – | + | / | * | -= | += | /= | *= | != | == | >= | > | <= | < |

Table 5:  Length (the number of tokens) statistics of buggy-HumanEval and buggy-FixEval.

| Percentiles | 50th | 90th | 95th | 98th | 99th | 100th |
|---|---|---|---|---|---|---|
| buggy-HumanEval | 214 | 470 | 550 | 595.6 | 606.7 | 617 |
| buggy-FixEval | 71 | 242.7 | 262 | 330 | 402.4 | 566 |

---

[5] https://github.com/openai/human-eval

[6] https://docs.python.org/3.9/library/ast.html

[7] We found 5 cases in our initial trials where the altered solutions still passed their test cases. Namely, changing the first applicable operators in the canonical solution of task IDs 'HumanEval/10', 'HumanEval/18', 'HumanEval/39', 'HumanEval/40', 'HumanEval/129'. Upon close examination, in these cases, either the altered solution is still correct or the altered solution is incorrect, but the test cases are too weak to tell.

```
"""Problem Statement: In the Kingdom of AtCoder, people use a language
called Taknese, which uses lowercase English letters.
In Taknese, the plural form of a noun is spelled based on the following rules:
If a nouns singular form does not end with s, append s to the end of the
singular form. If a nouns singular form ends with s, append es to the
end of the singular form.

You are given the singular form S of a Taknese noun. Output its plural form.
Input: Input is given from Standard Input in the following format:
S
Output: Print the plural form of the given Taknese word."""
```
Problem statement

```
s = input()
if s[len(s)-1] != s:
```
Partial code

(a) A prompt from buggy-FixEval, containing a problem statement and a buggy partial code.

```
s = input()
if s[len(s)-1] != "s":
    print(s + "s")
else:
    print(s + "es")
```
correct program

```
s = input()
if s[len(s)-1] != s:
    print(s + "s")
else:
    print(s + "es")
```
buggy program

(b) The original FixEval accepted (left) and rejected (right) user submissions.

Figure 4: **Example of a buggy-FixEval instance**. The example is based on `p02546`.

## A.2 Buggy-FixEval

FixEval[8] [19] is a dataset derived from CodeNet[9] [30] for studying program repair. It consists of programming problems from online programming contest websites[10], each with test cases, user-submitted solutions and judging verdicts to the submissions (*e.g., accepted* – passing all the tests; *wrong answer* – failing at least one test; or *time limit exceeded* – the program did not terminate within the specified time limit). The test cases in FixEval are more extensive than those in HumanEval, as the programming contest websites use them to automatically judge a submission's correctness. Each problem comes with an alphabetical label ("A" to "F") indicating the order in which the problem appears in its contest, with "A" usually being the easiest. Different from HumanEval, a solution or submission here is supposed to be a *complete* program, reading input from and writing output to the standard input and output instead of just a function implementation.

To derive a dataset for buggy-code completion, we pair each FixEval problem with its matching problem statements from CodeNet and discard problems without a match. Among the remaining problems, we focus on the relatively easy "A"-label problems because the post-"A" problems have been reported to be very challenging to solve even for recent language models of code: `pass@1` < 12% with CodeT5 [31] under a program-repair setting [19, Section 5.3].

We start by selecting rejected-accepted solution pairs. For each streak of Python submissions from a user to a problem, if the streak ends with an *accepted* submission, we take the last submission with the verdict being none of *accepted*, *compilation error* or *runtime error* as its rejected counterpart.

Then, we remove all comments and empty lines for each accepted-rejected pair and check the following. (1) Neither submission contains `exit()` or `system` calls, as they cause trouble for execution. (2) The first halves of the two submissions contain at least one non-white-space difference, including case difference, space/indent difference, or variable names for the assignment operator. (3) The character-level edit distance between the first halves of the two submissions is smaller than 20. Regarding the threshold of 20, we sampled 100 pairs of submissions with various edit distances. We qualitatively verified that under this threshold, the differences between the two submissions are, in the majority of cases, focused semantic changes (or likely bug fixes) and not refactoring or re-implementations.

---

[8]`https://github.com/mahimanzum/FixEval`
[9]`https://github.com/IBM/Project_CodeNet`
[10]The bases of the data are from AtCoder (`https://atcoder.jp/`) and AIZU Online Judge (`https://onlinejudge.u-aizu.ac.jp/`).

```
from typing import List
def has_close_elements(numbers: List[float], threshold: float) -> bool:
    """ Check if in given list of numbers, are any two numbers closer
    to each other than given threshold.
    >>> has_close_elements([1.0, 2.0, 3.0], 0.5)
    False
    >>> has_close_elements([1.0, 2.8, 3.0, 4.0, 5.0, 2.0], 0.3)
    True
    """
```

(a) The original HumanEval prompt containing a function header and a problem description as a docstring.

```
    for idx, elem in enumerate(numbers):
        for idx2, elem2 in enumerate(numbers):
            if idx != idx2:
                distance = abs(elem - elem2)
                if distance < threshold:
                    return True

    return False
```

(b) The canonical solution provided in HumanEval.

```
from typing import List
def has_close_elements(numbers: List[float], threshold: float) -> bool:
    """ Check if in given list of numbers, are any two numbers closer
     to each other than given threshold.
    >>> has_close_elements([1.0, 2.0, 3.0], 0.5)
    False
    >>> has_close_elements([1.0, 2.8, 3.0, 4.0, 5.0, 2.0], 0.3)
    True
    """
    for idx, elem in enumerate(numbers):
        for idx2, elem2 in enumerate(numbers):
            if idx == idx2:
                distance = abs(elem + elem2)
```

(c) The prompt used for buggy-code completion, containing the original problem and a buggy partial code.

```
def check(candidate):
    assert candidate([1.0, 2.0, 3.9, 4.0, 5.0, 2.2], 0.3) == True
    assert candidate([1.0, 2.0, 3.9, 4.0, 5.0, 2.2], 0.05) == False
    assert candidate([1.0, 2.0, 5.9, 4.0, 5.0], 0.95) == True
    assert candidate([1.0, 2.0, 5.9, 4.0, 5.0], 0.8) == False
    assert candidate([1.0, 2.0, 3.0, 4.0, 5.0, 2.0], 0.1) == True
    assert candidate([1.1, 2.2, 3.1, 4.1, 5.1], 1.0) == True
    assert candidate([1.1, 2.2, 3.1, 4.1, 5.1], 0.5) == False
```

(d) The test suite used for evaluating a completed program.

Figure 5: **Example of buggy-HumanEval instance.** The example is based on `HumanEval/0`. The buggy prompt (Figure c) is directly derived from the original prompt (Figure a) and the canonical solution (Figure b). We reuse the test suite (Figure d) for our evaluation.

The above process resulted in 492 pairs of prefixes from rejected-accepted solutions from 94 "A"-level problems. We manually inspected all the pairs to ensure the pairs satisfy the desiderata described in Section 3.1. Namely, we identify the following cases.

1. The prefix from the accepted solution is already a complete program that passes all the test cases. This can happen when the accepted solution is very short, for example, fewer than three lines, and/or followed by comments or unnecessary code. This counts for 26 pairs.

2. The prefix from the rejected solution and the prefix from the accepted solution are semantically equivalent up to variable name normalization. This counts for 33 pairs.

3. The prefix from the rejected solution contains no control flow (*e.g.,* `if`, `for`, `while`) and no output clause. In this case, a completion can ignore the prefix. This counts for 141 pairs.

4. The prefix from the rejected solution is bound to output wrong answers upon a certain input. For example, it contains a `print()` statement within a wrong condition. This counts for 25 pairs.

A list of identified cases can be found in our code repository. Case 1 makes the completion trivial from reference prefixes. Case 2 makes the comparison between the buggy completion and reference completion less meaningful. Case 3 makes the buggy completion less challenging. Thus, we exclude instances identified as cases 1, 2, and 3. We keep the case 4 instances because they fit the definition of potential bugs and are especially interesting in the sense that the "bugginess" already manifested in the code prefix.

After the above step, we are left with 292 pairs. We then use the first half of the rejected submission as a piece of buggy partial code and the first half of the accepted submission as the corresponding piece of reference code for buggy-code completion. A "buggy" prompt is formed by prepending the problem statement as a file-level docstring, or a multi-line string literal, to the buggy partial code. See an example in Figure 4.

# B   Details of rewriting-then-completion Method

With rewriting-then-completion, we attempt to eliminate potential bugs via a two-step procedure. We first locate the most likely line to contain a potential bug using a likelihood-based measure and then rewrite the line using an infilling code language model.

## B.1   Line Selection

Our underlying idea is to treat potential bugs as outliers in the generation flow of Code-LLMs. We observe that most reference code has lower perplexity than the corresponding buggy code.

**Likelihood-based measures.**   We calculate the score of each line as follows: With a model that can give a token distribution for each token location, we define the token score as the difference between the likelihoods of the most probable token (i.e., the `argmax` token) and the actual observed token. We get the score of a line by taking either the maximum or the average of all the non-zero token scores within it. The line with the largest score is finally selected to be rewritten.

To help better understand likelihood-based measures, consider the problem and the code prefix with potential bugs shown in Figure 8. The partial code has four lines in the function body. Given a language model $G$, for each token $x$, we calculate the token-level score as $p_2 - p_1$:

- $p_1$: probability of generating $x$ from the code up to $x$
- $p_2$: probability of generating $x^*$ from the code up to $x$, where $x^*$ gives the highest probability according to $G$.

For example, consider line 3 and token `==`. We would have $p_1 = 0.01$ (probability of $G$ generating `==`), and $p_2 = 0.95$ (`!=` is the most probable token according to $G$). The score of `==` is then 0.94. We obtain the line's score by taking the maximum scores at all token locations in the line. This way of aggregating token scores to line scores is referred to as "Likelihood (Max.)" Below is the variation used to report results in the main text.

Note that we take two measures to reduce the uncertainties of the likelihood estimation: (i) instead of the likelihood score of the target token itself, we use the maximal margin between the target token and the `argmax` token (similar to the popular approaches used in uncertainty quantification), and (ii) we aggregate the score gap along the line and set a high threshold to be more conservative.

We find that accuracies of localizing the line of potential bugs (with the same setting described above) are approximately 82% and 53% respectively on buggy-HumanEval and buggy-FixEval. For buggy-FixEval, we compare the detected line with the line of the first semantic difference between buggy and reference prefixes.

Table 6: `Pass@1` (↑) of each selection and completion method on the buggy-HumanEval and buggy-FixEval datasets. The best results for each dataset and completion model are in **bold**. The better likelihood aggregation method is underlined.

| Dataset | Selection Method | Completion Model | | | |
|---------|------------------|-------|------|-----------|-----------|
| | | CG-350M | CG-2B | INCODER-1B | INCODER-6B |
| buggy-HumanEval | Heuristic Oracle | **22.1** | **29.6** | **21.4** | **28.5** |
| | Likelihood (Max.) | 14.1 | 24.9 | 9.1 | 16.4 |
| | Likelihood (Avg.) | 13.2 | 23.0 | 9.8 | 16.9 |
| buggy-FixEval | Heuristic Oracle | 1.7 | 3.3 | 1.2 | 2.2 |
| | Likelihood (Max.) | 2.4 | 7.2 | **2.6** | 5.1 |
| | Likelihood (Avg.) | **2.7** | **7.9** | 2.4 | **5.4** |

**Heuristic oracle for comparison.** To see how well our likelihood-based measures work, we compare them against a heuristic oracle for predicting the line of potential bugs. In particular, we compare the buggy code prefix against the corresponding reference code prefix and select the first line with non-trivial differences. The differences are not about space, indent, or comment. Since this requires access to the corresponding reference code prefix, it cannot be used as part of a solution for bCC. We thus refer to it as an "oracle" and only use it here as a method of reference to gauge the performance of other line-selection methods. We do not use this oracle in any methods reported in the main text.

## B.2 Rewriting

After identifying the line most likely to contain a potential bug, we rewrite it by masking it and generating a replacement line using a code-infilling model such as INCODER [33].

## B.3 Results

Shown in Table 6 is our comparison between different likelihood-based measures. We observe that the Heuristic Oracle substantially outperforms the likelihood-based methods in buggy-HumanEval, but the reverse is true for buggy-FixEval. We provide a likely explanation for this phenomenon in the next paragraph. We also observe that Likelihood (Max.) outperforms Likelihood (Avg.) on average on buggy-HumanEval, but the reverse is true for buggy-FixEval. A possible explanation is that since the potential bugs in buggy-HumanEval take the form of a single token change, they are more amendable to being detected by focusing on the token with the largest score (maximum aggregation). In such a scenario, taking the average across all tokens in the line might dilute this signal. On the other hand, potential bugs are likely more subtle in a natural setting (represented by buggy-FixEval), and the potential bug may be spread out across multiple tokens. The fact that the average aggregation considers this may explain why it outperforms the maximum aggregation on buggy-FixEval.

We note that a large gap in performance between buggy-HumanEval and buggy-FixEval for the Heuristic Oracle method. This may be justifiable based on the information known to the Heuristic Oracle. In buggy-HumanEval, we introduce the bugs synthetically via a single semantically altering operator change. Hence, the oracle method has guaranteed knowledge of the line where the bug was introduced. This is not the case in buggy-FixEval since the examples were constructed directly from human submissions without synthetic perturbations. Therefore, the line corresponding to the first difference between these two submissions may not correspond to the bug location, or multiple (instead of one) locations can change between these two submissions. Therefore, this method is closer to a heuristic than a true oracle for buggy-FixEval. This is reflected in Heuristic Oracle's substantially weaker performance on FixEval. The fact that likelihood-based methods outperform the Heuristic Oracle on buggy-FixEval supports this explanation.

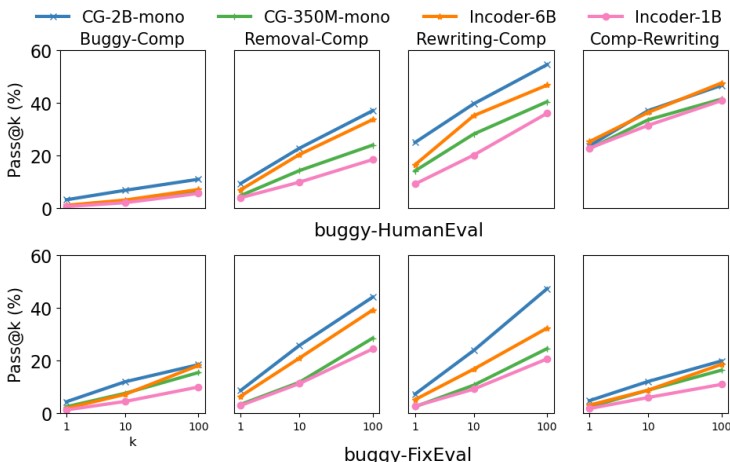

Figure 6: Comparison of CodeLMs per each completion method in terms of `pass@k`. CG: CODEGEN, Comp: Completion. We see the trend consistent with CODEGEN-2B-MONO > INCODER-6B > CODEGEN-350M-MONO > INCODER-1B across methods and $k$'s.

## C    Full Definition Statement and Additional Results

### C.1    Full Statement for Definition 2.1 in Section 2

**Definition C.1** (potential bug in a code prefix). Assume a problem given by specification $h$ with the test $f_h$ for evaluating its functional correctness and reference partial code $s$. Let $\mathcal{T}_s^h$ be the set of all valid programs, or *solutions* that satisfy $h$ and have the prefix $s$, *i.e.,* $\mathcal{T}_s^h = \{t := s :: c \,|\, f_h(t) = 1\}$. Let $\mathcal{C}_s^h$ be the set of all valid completed code, *i.e.,* $\mathcal{C}_s^h = \{c \,|\, s :: c \in \mathcal{T}_s^h\}$. A potential bug of $s$ is an edit $g_e$ on a token of $s$ that causes at least one existing solution to fail, *i.e.,* for $s' = g_e(s)$, there exists $c \in \mathcal{C}_s^h$ s.t. $f_h(s' :: c) = 0$. The prefix $s'$ is called *buggy* with respect to reference prefix $s$.

### C.2    Full Results with $k = 1, 10, 100$ and Interpretation for Section 4.3

Figure 6 presents the full comparison among `pass@k` for $k = 1, 10, 100$ for all methods. Intuitively, higher $k$ reveals more on the diversity of generated solutions. The relative performances between different Code-LLMs are consistent across different $k$'s. Regarding the relationship between completion methods,

- The naïve completion and completion-then-rewriting methods achieve relatively small increases as $k$ goes up, as the partial code constrains the input to the completion step.

- The increase in performance of removal-then-completion is relatively higher than naïve completion when $k$ increases, as its input is not constrained by the buggy partial code prefixes.

- The performance of rewriting-then-completion achieves relatively big increases among the methods as $k$ goes up. This is probably because rewriting-then-completion balances the completion-then-rewriting and removal-then-completion approaches. Thus, the pre-completion rewriting step increases the performance along with the diversity of code prefixes provided in the completion step.

### C.3    Performance on CODEGEN-16B-MONO

We provide results for CODEGEN-16B-MONO in table 7. As we can see, CODEGEN-16B-MONO also exhibits a similar phenomenon to our observations in previous Table 1 for smaller models, *i.e.,* potential bugs make Code-LLMs more challenging to generate correct programs and proposed mitigation methods still suffer from a large gap to the reference-prefix completion.

Table 7: `Pass@1` (↑) of completion methods with CODEGEN-16B-MONO. *Best method in each column is in **bold***.

| Prefix | Method | buggy-HumanEval | buggy-FixEval |
|---|---|---|---|
| reference | completion | 68.9 | 49.4 |
| buggy | completion | 4.4 | 8.0 |
| | removal-then-completion | 20.2 | **17.6** |
| | rewriting-then-completion | **24.3** | 9.7 |
| | completion-then-rewriting | 24.1 | 7.7 |

## C.4 Ablation Study: Concatenating Buggy-based Completion with Reference Prefix

In this ablation study, we use the buggy prefix to complete, then replace the buggy prefix with the reference prefix, shown in the bottom row of Table 8 (reference prefix + buggy-based completion). The results indicate that Code-LLMs fail to react to the change in the buggy prefix, as discussed in Section 4.2.

Table 8: Concatenating buggy-code-based completion with reference prefix.

| | buggy-HumanEval | | buggy-FixEval | |
|---|---|---|---|---|
| CODEGEN- | 2B | 350M | 2B | 350M |
| reference prefix + reference-based completion | 54.9 | 43 | 37.8 | 27.6 |
| buggy prefix + buggy-based completion | 3.1 | 0.7 | 4.3 | 2.4 |
| reference prefix + buggy-based completion | 40.8 | 31.6 | 4.6 | 2.5 |

## D Detailed Case Studies

For the case studies, We use the synthetic buggy-HumanEval dataset for our study. We use CODEGEN-2B-MONO as the completion model throughout this section as it achieved the best bCC performance overall. We surface interesting bCC examples by comparing pass rates for reference and buggy prefixes. For example, the Code-LLM may easily complete some reference prefixes (*i.e.,* high pass rate) but potentially fails with buggy prefixes (zero pass rate). Other examples may have a non-zero pass rate, meaning that the Code-LLM can adapt to the potential bug in the prefix and yield a correct solution. One observation that was observed uniformly about this study is that prefixes with potential bugs typically lead to lower pass rates, even when those were non-zero. This suggests that for these prefixes, the Code-LLM has to make more effort to find a correct solution.

### D.1 Are Potential Bugs Always Harmful?

As discussed in Section 2, potential bugs do not guarantee that the completed code will be buggy. These prefixes were sourced from completed examples that had a bug that could be ascribed to the prefix, but that does not mean by itself that the prefix will be impossible to complete correctly. While we observe the performance degradation of Code-LLMs under the presence of potential bugs, we find several interesting cases where these models generate correct code nonetheless.

Figure 12 shows that for the potential bug (highlighted operator) `==` modified from `!=` in the reference code, the completion model updates its original algorithmic flow with `continue` command and completes with correct code. While the new completion code is different from the canonical solution of the reference code, it is functionally correct. Thus, this implies that codellms may adapt in some recoverable cases.

### D.2 Why Do Code-LLMs Fail at bCC?

We investigate the cases where our models succeed with the reference partial code but fail with the buggy partial code.

## Problem specification

```
""" You're given a list of deposit and withdrawal operations on a bank account
    that starts with
zero balance. Your task is to detect if at any point the balance of account
    fallls below zero, and
at that point function should return True. Otherwise it should return False.
>>> below_zero([1, 2, 3])
False
>>> below_zero([1, 2, -4, 5])
True
"""
```

**Prefix with potential bugs**

```python
from typing import List
def below_zero(operations:
    List[int]) -> bool:
    balance = 0
    for op in operations:
        balance += op
        if balance >= 0:
```

**Prefix without potential bugs**

```python
from typing import List
def below_zero(operations:
    List[int]) -> bool:
    balance = 0
    for op in operations:
        balance += op
        if balance < 0:
```

**Model completion**

```python
            return True
        else:
            balance -= op
        if balance <= -1:
            return False
    return True
```

**Model completion**

```python
            return True
    return False
```

Figure 7: An example case of buggy-code completion where the model reacts to the code change but still fails. The bCC instance is based on `HumanEval/3`. The model completions shown here are representative completions for the given prefixes.

Our first finding is that the model often fails to react to the changes in code, *i.e.,* it produces the same completion for both buggy and reference code prefixes. We identify this by comparing if the most popular completions (out of all the sampled completions for a given instance) are the same for the reference and buggy code prefixes. We see that this happens in 90% of the instances and 93% of the problems (with at least one failed instance). Figure 7 shows an example. This suggests that the model is not sensitive to and thus ignores minor code changes and/or defaults to common patterns in the training data.

In cases where the most popular completions differ for the reference and the buggy code prefixes, we found that the potential bug often makes the code prefix significantly more challenging to complete correctly. In other words, the model may have recognized the potential bug and thus have significantly changed the output distribution but still failed. Figure 8 shows an example. Here, the balance check condition is reversed due to potential bugs. The model could ignore the given code or even define a new function to overwrite the buggy one. However, this drastically deviates from the commonly observed code patterns and thus may create a significant hurdle for the completion model if it is not introduced or adapted to such cases.

### D.3 When Do Code-LLMs Succeed at bCC?

Figure 9 shows an example when Code-LLMs ignore the if-else statement to bypass the potential bugs. In successful cases, we have observed that either the model ignores the incorrect state and generates the correct completion or considers the potential bug in the prefix to create a completion adapted to it.

```
""" Check if in given list of numbers, are any two numbers closer
to each other than given threshold.
>>> has_close_elements([1.0, 2.0, 3.0], 0.5)
False
>>> has_close_elements([1.0, 2.8, 3.0, 4.0, 5.0, 2.0], 0.3)
True
"""
```

**Prefix with potential bugs**

```python
from typing import List
def has_close_elements(numbers:
    List[float], threshold: float)
    -> bool:
    for idx, elem in
        enumerate(numbers):
        for idx2, elem2 in
            enumerate(numbers):
            if idx != idx2:
                distance = abs(elem -
                    elem2)
```

**Model completion**

```python
            if distance < threshold:
                return True
    return False
```

**Prefix without potential bugs**

```python
from typing import List
def has_close_elements(numbers:
    List[float], threshold: float)
    -> bool:
    for idx, elem in
        enumerate(numbers):
        for idx2, elem2 in
            enumerate(numbers):
            if idx == idx2:
                distance = abs(elem -
                    elem2)
```

**Model completion**

```python
            if distance < threshold:
                return True
    return False
```

Figure 8: An example case of bCC where the model fails to react to the code change. The bCC instance is based on `HumanEval/0`. The model completions shown here are the dominant completions for the given prefixes.

## D.4   How Are the Proposed Completion Methods Better at bCC?

The completion-then-rewriting method can detect and fix several types of suspicious operators. Figure 10 shows an example from the `string_xor` problem. Here, we see that the direct completion failed to react to the potential bug and continued as if the == operator was not changed to !=, which gives an incorrect solution. However, the code-repair model was able to identify and fix the != operator back to ==, producing a correct solution. Similarly, Figure 11 shows a success case from rewriting-then-completion, where it restores the introduced operator change. As buggy-HumanEval consists of bCC instances constructed from single operator changes, these methods can be particularly effective over naïve completion in circumventing the introduced operator change.

## Problem specification

```
""" Check if in given list of numbers, are any two numbers closer to each other
    than
given threshold.
>>> has_close_elements([1.0, 2.0, 3.0], 0.5)
False
>>> has_close_elements([1.0, 2.8, 3.0, 4.0, 5.0, 2.0], 0.3)
True
"""
```

**Prefix with potential bugs**

```python
from typing import List
def has_close_elements(numbers:
    List[float], threshold: float)
    -> bool:
    for idx, elem in
        enumerate(numbers):
        for idx2, elem2 in
            enumerate(numbers):
            if idx == idx2:
                distance = abs(elem -
                    elem2)
```

**Prefix without potential bugs**

```python
from typing import List
def has_close_elements(numbers:
    List[float], threshold: float)
    -> bool:
    for idx, elem in
        enumerate(numbers):
        for idx2, elem2 in
            enumerate(numbers):
            if idx != idx2:
                distance = abs(elem -
                    elem2)
```

**Model completion**

```python
                continue
            else:
                distance = abs(elem -
                    elem2)
                if distance < threshold:
                    return True
    return False
```

**Model completion**

```python
                if distance <
                    threshold:
                    return True
    return False
```

Figure 9: An example case of bCC where the model bypasses the potential bug and succeeds. The bCC instance is based on HumanEval/0. The model completion for the reference prefix is the most common. The model completion for the buggy prefix is one of the two successful completions.

```
""" Input are two strings a and b consisting only of 1s and 0s.
Perform binary XOR on these inputs and return result also as a string.
>>> string_xor('010', '110')
'100'
"""
```

**Prefix with potential bugs**

```
from typing import List
def string_xor(a: str, b: str) ->
    str:
    def xor(i, j):
        if i != j:
            return '0'
```

**Model completion**

```
        else:
            return '1'

    return ''.join([xor(i, j) for i,
        j in zip(a, b)])
```

**Output from completion-then-rewriting**

```
from typing import List
def string_xor(a: str, b: str) ->
    str:
    def xor(i, j):
        if i == j :
            return '0'

        else :
            return '1'

    return ''.join([xor(i, j) for i,
        j in zip(a, b)])
```

Figure 10: An example case of bCC where the completion-then-rewriting method successfully fixes the potential bug. The bCC instance is based on `HumanEval/11`. The model completion for the buggy prefix is representative.

```
""" Check if in given list of numbers, are any two numbers closer to each other
    than
given threshold.
>>> has_close_elements([1.0, 2.0, 3.0], 0.5)
False
>>> has_close_elements([1.0, 2.8, 3.0, 4.0, 5.0, 2.0], 0.3)
True
"""
```

**Prefix with potential bugs**

```
from typing import List
def has_close_elements(numbers:
    List[float], threshold: float)
    -> bool:
    for idx, elem in
        enumerate(numbers):
        for idx2, elem2 in
            enumerate(numbers):
            if idx == idx2:
```

**Rewritten prefix**

```
from typing import List
def has_close_elements(numbers:
    List[float], threshold: float)
    -> bool:
    for idx, elem in
        enumerate(numbers):
        for idx2, elem2 in
            enumerate(numbers):
            if idx != idx2 and
                abs(elem - elem2) <
                threshold:
```

**Model completion**

```
                return True
    return False
```

Figure 11: An example case of bCC, where the rewriting-then-completion method successfully fixes the potential bug. The bCC instance is based on `HumanEval/0`.

```
Check if in given list of numbers, are any two numbers closer
to each other than given threshold.
```

```python
from typing import List
def has_close_elements(numbers: List[float],
                       threshold: float) -> bool:
    for idx, elem in enumerate(numbers):
        for idx2, elem2 in enumerate(numbers):
            if idx != idx2:

                distance = abs(elem - elem2)
                if distance < threshold:
                    return True
  return False
```

```python
from typing import List
def has_close_elements(numbers: List[float],
                       threshold: float) -> bool:
  for idx, elem in enumerate(numbers):
        for idx2, elem2 in enumerate(numbers):
            if idx == idx2:

        continue
            if abs(elem - elem2) < threshold:
                return True
    return False
```

Without potential bugs        With potential bugs

Figure 12: **A success case of buggy-code completion**. The model manages to generate a functionally correct completion by surpassing the potential bug (highlighted) by a `continue` command. The completions are from CODEGEN-2B-MONO on `HumanEval/0` from buggy-HumanEval.

