# OpenReview forum: "Large Language Models of Code Fail at Completing Code with Potential Bugs"
_NeurIPS.cc/2023/Conference — NeurIPS 2023 poster_

### Official Review · Reviewer_8UBb · 2023-06-20

**Soundness:** 2 fair
**Presentation:** 3 good
**Contribution:** 2 fair
**Rating:** 7
**Confidence:** 4

**Summary:**

In general, the code completion task that is commonly used to evaluate LLMs assume a clean, non-buggy starting point. However, in real-world software development, this assumption is not always valid. In this papers, the authors investigate code completion given a buggy prefix. For this, they derive two benchmarks from existing datasets. Namely, they build buggy-HumanEval by introducing semantics-altering operator changes in the HumanEval dataset and buggy-FixEval based on user submissions in coding contests. They find that performance is lower in the presence of a buggy prefix, and they introduce different techniques for mitigating this to some extent.


Contributions:
- Evaluate a handful of Code LLMs on code completion with a buggy prefix.
- Introduce two benchmarks that will be publicly released.
- Investigated different post-hoc mitigation techniques which demonstrate some improvement in performance.

**Strengths:**

- The authors introduce two useful datasets, with synthetic and real examples, that could serve as benchmarks of buggy code completion for the research community.
- The task of buggy code completion is quite interesting and also important, though it may not be completely novel, as there is prior work (“Learning to Extend Program Graphs to Work-in-Progress Code”) outside the context of LLMs that studies a similar setting.
- The post hoc mitigation techniques are useful in understanding the current limits.
- There are nice ablation studies and qualitative analysis with interesting findings. Specifically, it’s cool to see that sometimes the completion changes to adapt to the potential bug.


**Weaknesses:**

- To run the test suite, both the prefix and predicted suffix are needed. The bug is in the prefix that is given to the model (not generated by the model). So, it is possible that tests fail only because of the bug in the prefix and not because of anything the model generated. In lines 96-101, the authors explain that they loosen the constraints such that they allow the model to make changes to the prefix; however, this is an additional bug fixing task that the model needs to perform, on top of code completion. In other words, in the buggy code completion setting, there are two possible ways to pass the tests. (1) Fix the bug in the prefix and then complete or (2) Try to complete the code by adapting to the buggy code. Therefore, it doesn’t seem fair to compare directly with completion only based on a clean prefix. Since the main claim of this paper is that the task of code completion suffers as a result of buggy prefixes, I feel that it is important to evaluate the completion component alone. Would it be possible to decouple things a bit? For example, when running the test suite only, would it be possible to replace to buggy prefix with the clean prefix to evaluate whether the completion would have been correct if the prefix (which is not model generated) was correct?
- For the post-hoc mitigation techniques, the rewriting uses a BERT-based model which is a considerably weaker model, and even though some code pre-training is done on top of it, it likely does not reach the level of modern code LLMs. Perhaps using a stronger model would have yielded better performance.

**Questions:**

- Please clarify the point I raised in the section above.
- The authors claim multiple times that code repair on short context is an ill-defined problem. No references are provided here, and actually, people have studied such tasks in the literature (e.g., bug-fixing with a single statement such as ManySStuBs4J or handful of statements such as the Tufano et al., BFP dataset). Could the authors provide an explanation for their argument?

- When introducing new benchmarks, it is good to provide dataset statistics (e.g., average lengths).
- Missing reference: “Learning to Extend Program Graphs to Work-in-Progress Code” (they consider code completion on work-in-progress code) [Li, Maddison, and Tarlow; 2021]
- Consider placing Figure 2 closer to the results.
- It would be interesting to run your experiments on GPT-3.5 or GPT-4.


**Limitations:**

One limitation that the authors have not mentioned is the assumption that FixEval closely aligns with a real-world scenario. Coding contest problems do not necessary align with a more general software engineering setting. However, it is challenging to get a test suite for such a setting and so it is more difficult to evaluate.

---

> ### Author Rebuttal · Authors · 2023-08-10
>
> We thank the Reviewer for the detailed review and constructive suggestions.
> We are encouraged that the Reviewer finds our task interesting and important with our studies and analysis providing interesting and cool findings.
> We appreciate your acknowledgment of the thorough evaluation of our insights as well as the contribution of new benchmark datasets and post hoc mitigation techniques.
> Please find our answers to your comments and questions as follows.
>
>
> **Comment:** *It is possible that tests fail only because of the bug in the prefix and not because of anything the model generated.*; *it doesn’t seem fair to compare directly with completion only based on a clean prefix.*; *Would it be possible to decouple things a bit?*
>
> Thanks for your suggestion.
> First, we note that:
> * The performance with “clean prefix” is considered a reference to see how potential bugs affect the LLM rather than a direct method for comparison, so our purpose is to measure the gap between scores when using buggy and clean prefixes to relate how well models react to the change.
> * The “potential bugs” in buggy prefixes do not necessarily make the model fail. Instead, a good model should manage to complete buggy prefixes with a suitable completion to satisfy the test cases. We showed these interesting examples in Section 4.6.
>
> Per your suggestion, we have evaluated and reported the following results (pass@1 score):
>
> |buggy-| HumanEval|HumanEval | FixEval| FixEval|
> |:---:|:---:|:---:|:---:|:---:|
> |codegen- | 2B-mono|350M-mono|2B-mono|350M-mono|
> |Clean prefix + clean-based completion| 54.9|43 | 37.8|27.6 |
> |Buggy prefix + buggy-based completion| 3.1| 0.7| 4.3| 2.4|
> |Clean prefix + clean-based completion| 40.8| 31.6| 4.6| 2.5|
>
> Here, the bottom row (Clean prefix + clean-based completion) is when we use the clean prefix and the completion of the buggy prefix. As we can see, the result reflects that models fail to react to the change in the buggy prefix, as discussed in Section 4.2 (*Why do Code-LLMs fail at bCC?*).
>
> **Comment:** *For the post-hoc mitigation techniques, the rewriting uses a BERT-based model which is a considerably weaker model ...  Perhaps using a stronger model would have yielded better performance.*
>
> In the rewriting method, we use Incoder [Fried et al., 2022] because, to the best of our knowledge, it is the latest and the best LLM trained for infilling tasks — suitable for fixing bugs in the middle of a program.
> In fact, Incoder is both (left-to-right generation) as well as editing (via masking and infilling).
> Nevertheless, our design allows us to easily replace Incoder with a better infilling model in the future for better repairing performance.
>
> *[Fried et al., 2022]. Incoder: A generative model for code inﬁlling and synthesis. arXiv preprint arXiv:2204.05999, 2022
>
>
> **Q:** *The authors claim multiple times that code repair on short context is an ill-defined problem. … Could the authors provide an explanation for their argument?"*
>
> Thanks for your question. In fact, our statement is that bugginess is only meaningful and a property of a **complete program**; thus (in line 31), we meant that given **the incomplete and short** code context* (i.e., without a full program), it’s ill-deﬁned for detecting or fixing bugs.
> Thanks for your reference, In [Tufano et al., 2019], we believe while the context is short, it’s a full program, e.g., JAVA programs in Figure 2.
> We will revise the statement to make it transparent.
>
>
> **Q:** *When introducing new benchmarks, it is good to provide dataset statistics (e.g., average lengths).*
>
> Thanks. We provided some detailed statistics of datasets in the supplementary (Section A).
> Per your suggestion, we also report the length statistics (in terms of the number of tokens) as follows.
>
> |Percentiles|50th|90th|95th|98th|99th|100th|
> |:---:|:---:|:---:|:---:|:---:|:---:| :---:|
> |buggy-HumanEval|214|470|550|595.6|606.7|617|
> |buggy-FixEval|171|242.7|262|330|402.4|566|
>
> We will add more statistics in the final version.
>
> **Q:**  *Missing reference: “Learning to Extend Program Graphs to Work-in-Progress Code” (they consider code completion on work-in-progress code) [Li, Maddison, and Tarlow; 2021]*
>
> Thanks for bringing this work to our attention. We note that this work focuses on the transfer learning of token-relation prediction for code completion under work-in-progress code scenario, different from our setting and no-training post-hoc techniques.  We have added this work to the related work.
>
> **Q:**  *It would be interesting to run your experiments on GPT-3.5 or GPT-4.*
>
> Thanks for your suggestion. Due to the limited time and flexibility, we have conducted extra experiments with a very large model CodeGen-16B-mono instead of GPT-4.
> We are also planning to run our experiment on GPTs and incorporate results in the next version.
>
> Here are the results on CodeGen-16B-mono
> |buggy-HumanEval (200 instances) | pass@1| pass@10|pass@100|
> |:---:|:---:|:---:|:---:|
> |Clean prefix| 78.5 | 93.1 | 95.7 |
> |Buggy prefix| 8.9 | 20.4 | 25.2|
> |Removal| 43.2 | 70.9 | 80.1 |
> |Completion->rewriting| 14.4 | 29.7 | 35.6|
> |Rewriting->completion|70.9 | 80.2 | 86.9 |
>
> |buggy-FixEval (100 instances)| pass@1| pass@10|pass@100|
> |:---:|:---:|:---:|:---:|
> |Clean prefix| 59.7 | 72.6 | 75.7 |
> |Buggy prefix| 8.5 | 18.8 | 37.7 |
> |Removal| 20 | 49.8 | 68 |
> |Completion->rewriting| 8.6 | 18.4 | 37.7|
> |Rewriting->completion|8.9 | 19 | 32.9 |
>
> As we can see, CodeGen-16B-mono exhibits a similar phenomenon to our observations.
>
> **Minor**:
> Thanks. We revised.
>
> **Limitation:** *Coding contest problems do not necessary align with a more general software engineering setting*
>
> Thanks. We will elaborate on this limitation in our manuscript.
>
> ---
> **Final notes:** We hope our responses and new experiment results can help you better appreciate our work and consider increasing your score and supporting accepting our paper. Thanks again for providing valuable feedback!

---

> > ### Comment · Reviewer_8UBb · 2023-08-15
> >
> > Thank you for providing detailed explanations and new results. Just a few more points:
> >
> > - In the first table you provided, why are the results for FixEval surprisingly low for clean prefix + buggy-based completion, compared to HumanEval?
> > - For the last two CodeGen tables, could you also include this clean prefix + buggy-based completion? From the HumanEval results in the first table, it seems like replacing the buggy prefix with the clean prefix helped to significantly close the gap in performance for HumanEval. I'd like to see how that affects stronger models as well.

---

> > > ### Author Response · Authors · 2023-08-16
> > > **Re: Official Comment by Reviewer 8UBb**
> > >
> > > Thank you for your comments. Please find our answers as follows,
> > >
> > > **Question:** *In the first table you provided, why are the results for FixEval surprisingly low for clean prefix + buggy-based completion, compared to HumanEval?*
> > >
> > > In buggy-FixEval (bFE), a buggy prefix may be very different from its corresponding clean prefix, e.g., more than one place of differences, variable changes, etc., while in buggy-HumanEval, the difference between a buggy and a clean prefix is guaranteed to be a single operator change. This means that a code language model is more likely to generate different completions for a buggy prefix in bFE than for one in bHE. Thus, simply concatenating the clean prefix with a model completion from the buggy prefix is less likely to result in a functional program for bFE.
> > >
> > > **Question:** *For the last two CodeGen tables, could you also include this clean prefix + buggy-based completion? … I'd like to see how that affects stronger models as well.*
> > >
> > > Here we include “clean-prefix + buggy-based completion” pass@1 results using CodeGen-16B-mono.
> > >
> > > |buggy-| HumanEval (200 instances)  | FixEval (100 instances)|
> > > |:---:|:---:|:---:|
> > > |Clean prefix + clean-based completion| 78.5 | 59.7|
> > > |Buggy prefix + buggy-based completion| 8.9 | 8.5|
> > > |Clean prefix + buggy-based completion| 49.4| 10.6|
> > >
> > > We see that the trend is consistent where CodeGen-16B-mono is able to reach a relatively high pass rate compared to clean prefix completions on buggy-HumanEval, but not so well on buggy-FixEval.
> > >
> > > **Comment:** *It seems like replacing the buggy prefix with the clean prefix helped to significantly close the gap in performance for HumanEval.*
> > >
> > > We would like to remark that a high pass rate on bHE suggests more about a model’s inability to adapt to the potential bugs injected into the prefix rather than the difficulty of the buggy prefix itself. In particular, a success case for "clean prefix + buggy-based completion" is almost always a failure case for "buggy prefix + buggy completion", as is guaranteed by the tested semantic-altering operator flip in the prefix part.
> > > Meanwhile, the failure of all "buggy prefix + buggy-based completion" cases for a buggy prefix does not imply that a functionally correct buggy-based completion is impossible. Instead, it suggests the model’s inability to come up with such completions.
> > >
> > > Nevertheless, while access to clean prefixes is unavailable in our setting (only for the relative comparison), the "clean prefix + buggy-based completion" results shed insight into the counterfactual performance of buggy code completion and help quantitatively demonstrate the current models' insensitivity to adapting to potential bugs.
> > >
> > > Thanks again for the great questions and suggestions. We will incorporate this discussion into our revised version.

---

> > > > ### Comment · Reviewer_8UBb · 2023-08-17
> > > >
> > > > Thank you for the detailed answers to my questions. I have increased my score to 7.

---

> > > > > ### Author Response · Authors · 2023-08-19
> > > > > **Re: Official Comment by Reviewer 8UBb**
> > > > >
> > > > > Thank you for the positive feedback!

---

### Official Review · Reviewer_Cn5Q · 2023-07-06

**Soundness:** 2 fair
**Presentation:** 4 excellent
**Contribution:** 2 fair
**Rating:** 4
**Confidence:** 4

**Summary:**

The paper studies the ability of large language models for code to handle prompts which contain “buggy” code. To this end, it introduces two datasets: (i) buggy-HumanEval, which is constructed by injecting synthetic wrong binary operator bugs into HumanEval problems, and (ii) buggy-FixEval, which is constructed from real bugs in user submissions to competitive programming tasks. The evaluation finds that “buggy” prompts significantly degrade LLMs’ performance. The paper also investigates mitigation methods for “buggy” prompts.

**Strengths:**

The strengths of the paper are as follows:
1. The paper is well-written and easy-to-follow. It provides useful examples for illustration.
2. The paper introduces two new datasets, which can be useful for future research.
3. The evaluation is thorough and provides new insights.


**Weaknesses:**

### Main weaknesses
Definition 2.1 is a core definition in the paper but has two major issues. First, defining bugginess over code prefixes sounds incorrect to me, because bugginess is a property for complete programs. Second, the original prefix $s$ can also be viewed as a “buggy prefix”: there likely exists $c’$ such that $t’ = s’ :: c’$ satisfies $h$ and $t = s :: c’$ does not satisfy $h$. In fact, the paper shows the existence of such cases (e.g., Figure 12).

Consider a “buggy prefix” $s’$ and a program $t’ = s’ :: c’$. Under what condition exists $c’$ such that $t’$ satisfies $h$? Or when is $t’$ unlikely to satisfy $h$? For these two questions, the paper provides some empirical evidence (e.g., Figure 12 and Lines 640-642) but lacks a theoretical analysis.

A highlight of the paper is the discussion over synthetic and real bugs (buggy-HumanEval vs. buggy-FixEval). However, the paper does not capture data imbalance in real bug distribution. That is, bugs occur very infrequently in practice, as shown in [1] and [2] below (not discussed in the paper). Ignoring such data imbalance can threaten the validity of some results. For example, Tables 2 and 3 are obtained on datasets with a balanced ratio of “buggy” and “non-buggy” prefixes.

- [1] He and Vechev, On Distribution Shift in Learning-based Bug Detectors, ICML 2022.
- [2] Karampatsis and Sutton, How Often Do Single-Statement Bugs Occur? The ManySStuBs4J Dataset, MSR 2020.

For buggy-FixEval, the absolute improvement from rewriting->completion or completion->rewriting is small, suggesting that these methods might be of limited practical effectiveness.

### Other questions:
1. In Table 1, the performance of removal->completion on buggy-HumanEval is very low. This is surprising to me, because I would imagine the performance to be much better, if the original HumanEval prompt is used as input (as in Figure 5(a)). How do you perform the removal exactly?
2. In Section 4.3, the results of completing “clean prefixes” are treated as the upper bounds for completing “buggy prefixes”. This does not sound correct to me. Again, I think the correct upper bound is 100 minus the percentage of “buggy prefix” $s’$ for which no program $t’ = s’ :: c’$ satisfies $h$.

### Small issues:
1. Line 235: I could not find Appendix C.3.
2. Line 665: figure 8 -> Figure 8.
3. Table 5 is not referenced in Appendix B.3.


**Questions:**

Please consider addressing the points raised in the “Weakness” section.

**Limitations:**

The paper provides a sufficient discussion of limitations and potential negative impacts.

---

> ### Author Rebuttal · Authors · 2023-08-10
>
> We thank Reviewer Cn5Q for the detailed review and constructive suggestions.
> We are encouraged that the Reviewer find our paper well-written and easy to follow with illustrated examples.
> We appreciate your acknowledgment of the thorough evaluation of our insights and the contribution of new benchmark datasets.
>
>
> **Comment:** *bugginess is a property for complete programs.*
>
> Thanks. We completely acknowledged this property of bugs in the manuscript that “bugginess” is a property of complete programs, not code prefixes (and the bugginess of a code prefix can only be assessed in conjunction with a completion.)
> Therefore, our work referred to the studied patterns as "potential bugs", not “bugs” per see.
>
> For brevity, we sometimes drop “potential” or “potentially” when mentioning potential bugs and potentially buggy prefixes when the context is clear. We will make this point clearer near the definition.
>
>
> **Comment:** *The original prefix can also be viewed as a “buggy prefix”....  For these two questions, the paper provides some empirical evidence ... but lacks a theoretical analysis.*
>
> Thanks for a great comment.
> Generally speaking, the bugginess level of a prefix is relative to w.r.t. a reference prefix; thus, our definition means a prefix potentially buggy is w.r.t. a  reference prefix (which we name a “clean” prefix). Specifically, “potential bug” as some pattern occurring unnaturally w.r.t to a naturally written program.
> However, under current definition in manuscript, it is indeed a possibility that the original prefix can also be viewed as a “buggy prefix”.
> We will refine these terms (clean as reference prefix, and buggy as relative/potential buggy).
> Here, we provide a detailed clarification and hope this helps Reviewer better assess our intuition.
> * Intuitively, the reference program t:= s::c captures the natural distribution of programs that correlate s to preferred implementation for certain programming tasks, e.g., functionally correct, easy to read, follows software development best practices, etc. So, it’s likely that deviations from it can be probably bad, especially when they introduce functional incorrectness.
> * A more holistic definition would be to quantify naturalness and various aspects of preferred quality traits and define the “potential bugginess” as a likelihood value. However, quantifying the property and distribution of counterfactual implementation t’ and completion c’ is inherently difficult, if not impossible. Even full-scale empirical analysis would require significant effort in collecting and reviewing cases manually. We provide empirical observations with model completions.
>
> We go for the current operational definition for its simplicity and verifiability. It suffices to guide us through the initial exploration of the impact of this previously under-investigated scenario.
> Nevertheless, we will refine the terminologies (buggy, clean) to reflect our definition and intuition precisely.
>
> Regarding the analysis, our work aims to answer these two questions empirically first for further theoretical analysis in future studies.
>
>
>
>
> **Comment:** *Imbalance in real bug distribution: Ignoring such data imbalance can threaten the validity of some results.*
>
> Thanks. Our work identifies and focuses on studying the behavior of LLMs in a setting with potential bugs. How often do potential bugs appear in actual code development when a user triggers auto-completion is indeed an open question and worth studying. It is however beyond the scope of this study. We provide preliminary observations on balancing the performance on potentially buggy and clean prefixes in Tables 2 and 3. We hope our study will raise awareness about the existence of potential bugs and call for a method that deals well with whether the potential bug exists or not.
>
> **Comment:** *methods might be of limited practical effectiveness*
>
> Thanks. Our proposal for these simple strategies is to help understand the limits of current LLMs instead of an advanced technique.
> We hope our study can motivate further study of advanced methods.
>
> **Q:** *How do you perform the removal exactly?*
>
> For the *removal* method, we remove the partial code from the prefix to keep only the problem statement and the function header (with optional examples), so the input of removal is as same as the prompt in the original HumanEval.
> We note that:
> * We double-checked that on the HumanEval, our removal method reproduces the pass@k performance of Codegen models reported in [Nijkamp et al., 2022].
> * buggy-HumanEval is derived from a subset of considerably longer and harder problems in HumanEval. In particular, the average number of lines of our selected problems is 8.2 compared to 2.9 of the rest, resulting in much lower pass@k scores in our subset.
>
>
>
> **Q:** *The correct upper bound is 100 minus the percentage of “buggy prefix” s' for which no program t' = s'::c' satisfies h*
>
> Thanks. We agree that the term "upper bound" here is improperly used. We will refine the term 'upper bound' to “expectation from reference prefixes”'.
> While the suggested bound is theoretically the upper bound, an empirical estimate seems not plausible because of the number of such buggy s'.
> Thus, we use the results on the "clean prefix" as expected to evaluate how well the model can achieve with the corresponding "buggy prefix".
>
> **Minor:**: Thanks. We fixed these issues.
>
> ---
> **Final note:** We want to thank you again for such exemplary and detailed comments. We will integrate these answers into our new version. We hope our responses can help you better appreciate our work and consider increasing your score and supporting accepting our paper. We agree that our current definition has some ambiguity though it serves the purpose of empirical verification. Still, with its timely development, novelty, significance, and potential in future research, we believe our work can have a good impact on the research society. Thanks again for your careful reading!

---

> > ### Comment · Reviewer_Cn5Q · 2023-08-14
> >
> > I have read other reviews and the author rebuttals. I would like to thank the authors for providing helpful clarifications. However, I think two fundamental issues are not fully addressed, as discussed below. I don’t see how they can be addressed without major rewriting. Therefore, I still consider this paper borderline and do not raise my rating.
> >
> > ### Definition
> >
> > I still don’t find “potential bugs” a reasonable term because LLMs can potentially generate bugs from any prefix. I think (as also suggested in the author rebuttal), the property studied in this paper should be the “naturalness” of the prefix, in the context of functional correctness for code completion. That is, LLMs are accurate at generating functionally correct code for natural prefixes (e.g., the original prefixes in HumanEval), but fail to generalize to unnatural prefixes (e.g., those with anti-patterns applied).
> >
> > ### Data imbalance
> >
> > I understand the results in Tables 2 and 3. But they are done in an unrealistic setting with a balanced ratio of “buggy” and “clean prefixes”. In a real-world setting, it is critical to consider an imbalance buggy-clean ratio (e.g., 1:1000 as shown in the paper I cited). Due to the large number of “clean” prefixes, even a slight decrease of pass@1 on “clean” prefixes will drastically increase the absolute number of bugs. While your approach improves pass@1 for “buggy” prefixes, it might only translate to a minor reduction in the absolute number of bugs. As a result, the approach might end up with introducing more bugs.
> >
> > I suggest the authors to at least cite the two papers I mentioned and provide a discussion on this, such that the readers can have a correct understanding on the practical usefulness of your approach.

---

> > > ### Author Response · Authors · 2023-08-15
> > > **Re: Official Comment by Reviewer Cn5Q**
> > >
> > > Thanks for your prompt response. Please find our response as follows,
> > >
> > > ### **Data imbalance**
> > > Thanks for the suggestion. We have carefully read the papers and added them to our manuscript. Due to the limited length of the rebuttal this year, we couldn’t present our discussion. Here is our  detailed discussion on related works:
> > > > [1] finds that distribution shifts between distributions of artificial and real bugs make trained models unusable in practice (<10% precision). Thus, [1] exposes and attempts to mitigate the negative impact of bug repairers caused by the domain shift between synthetic training bugs and realistic test bugs with a proposed two-stage training approach: first on a balanced synthetic dataset, then on an imbalance data with a small number of real bugs.  It is relevant to us in 1) that we access the naive completion and baseline mitigations on both synthetic and realistic buggy-code completion benchmarks, and 2) that there is possibly a similar impact of domain shift for the program rewriters used in our baseline methods, as they may not have been trained over such examples. [2] focuses on single-statement bug fixes. They reported an interesting data point on how prevalent single-statement bug fixes appear in popular open-source Java projects. There are more bugs beyond “simple bug fixes”. Finished code in popular open-source projects is likely to have much higher quality than work-in-progress draft code. There may be unidentified bugs. The bug density may be different between Java and Python. Thus, the presence of potential bugs can be much more frequent.
> > >
> > > * [1] He, Jingxuan, Luca Beurer-Kellner, and Martin Vechev. "On distribution shift in learning-based bug detectors." In International Conference on Machine Learning, pp. 8559-8580. PMLR, 2022.
> > > * [2] Karampatsis, Rafael-Michael, and Charles Sutton. "How often do single-statement bugs occur? the manysstubs4j dataset." In Proceedings of the 17th International Conference on Mining Software Repositories, pp. 573-577. 2020.
> > >
> > > In the revised version, we will add a discussion on the practical usefulness of real-world imbalanced bug distribution.
> > >
> > > *However*, it’s worth noting that our work is more orthogonal and complements the existing (suggested) works:
> > > 1. **Scope of our paper:** Our main point of the paper is to expose the issue and study the impact when potential bugs are present. The baseline methods are first attempts to demonstrate the task's difficulty rather than offer a well-rounded solution that works in all cases. Also, the current assessment can tell us how a method deals with buggy prefixes relatively well.
> > >
> > > 2.  Beyond our scope, how can we use our provided datasets and evaluation for imbalanced settings?
> > > * (i) *How about the current evaluation?*
> > > > One can use the weighted version of pass@k between clean and buggy prefixes, given prior knowledge about the clean/bug ratio.
> > > * (ii) *Regarding our methods,* the distribution shift [1] more or less affects the train-based approaches significantly, as mentioned in [1], while our approaches are post-hoc, without any training
> > > * (iii) Furthermore, one can use our balanced dataset for training. For instance, in [1], the approach can use our balanced dataset for the first stage in their two-stage approach. Given prior knowledge about the clean-buggy ratio, one can sample from our clean and buggy datasets to create an imbalanced dataset.
> > >
> > > ### **Potential Bug Definition**
> > > As mentioned in our response, we will revise the definition to make it more precise with more elaboration on the meaning. Our team debated using "naturalness" or "potential bugs" to describe this phenomenon. Eventually, it settled on "potential bug" since we believe this to be more descriptive of this phenomenon than "naturalness," which would be even more ambiguous.
> > > Furthermore, while the terms can be refined, they don’t significantly affect the rest of the paper and results.
> > >
> > > ---
> > > Given these two points, we will refine the definition of potential bugs and the discussion on imbalance in the final revision. We hope that our responses will further address your concerns.

---

> > > > ### Author Response · Authors · 2023-08-19
> > > > **Re: Official Comment by Reviewer Cn5Q**
> > > >
> > > > Dear Reviewer Cn5Q,
> > > >
> > > > We deeply value the time and effort you've spent reviewing our work. We are happy to offer further clarification or feedback if there are any other questions or concerns.
> > > >
> > > > Thanks.

---

> > > > > ### Comment · Reviewer_Cn5Q · 2023-08-21
> > > > >
> > > > > Thank you for your response. However, I believe that my score is still a fair assessment.

---

### Official Review · Reviewer_h8sG · 2023-07-06

**Soundness:** 3 good
**Presentation:** 3 good
**Contribution:** 3 good
**Rating:** 7
**Confidence:** 4

**Summary:**

This paper identifies and analyses a limitation of the LLMs for code, that they fail for completing code with potential bugs. This is a problem because LLMs are deployed in real world software projects which may contain many bugs. The authors attempt to bridge this gap between test environment and real world by creating buggy versions of two popular datasets used for testing LLMs code generation capabilities. They experiment with two models, with two sizes each and show how their performance is impacted by introducing potential bugs in the test dataset. The authors also try different techniques of addressing the limitations of LLMs when trying to complete potentially buggy code.

**Strengths:**

The authors have highlighted a relevant problem with LLMs. It is a well written paper and the authors demonstrate a good understanding of the problem. They consider many different possibilities when generating the data and run thorough experiments covering many scenarios. They also explore intuitive solutions to overcome the limitation of LLMs that they have highlighted.
The strengths of the paper are:
1. The paper highlights an interesting and a very real problem.
2. The suggest good post-hoc strategies and cover some of the gap in the model performance introduced by the buggy dataset.
3. They experiment with multiple datasets.
4. They experiment with multiple open source models of different sizes.
5. They will make the new buggy version of the dataset available to public.

**Weaknesses:**

The paper has the following weaknesses:
1. The scope the problem highlighted is very narrow. It is limited to code generation, and even within code generation it is limited to code completion. It does not even apply to natural language to code generation. This is completely because of the nature of the problem highlighted.
2. They have experimented with multiple models of different sizes, but they did not try a very large LLM. As we know that as the model size increases, new capabilities get added to the model. This leaves the question open as to whether a much larger model, like 16B parameter CodeGen, would exhibit the same limitation or have the same gap in performance after post-hoc techniques.

**Questions:**

Do you think a much larger model will have the same limitation? Do you plan to experiment with larger models?

For code completion applications deployed in IDEs, rewriting user code is generally not expected. How does this limitation, and the fact (which you have mentioned in the paper) that it may be impossible for the model to avoid some potential bugs, affect the scope of the problem?

More generally, how do you expect code completion applications to deal with buggy code?

**Limitations:**

The authors should write more about the scope of the problem and it's impact on code completion applications.

---

> ### Author Rebuttal · Authors · 2023-08-10
>
> We thank the Reviewer for the detailed review and constructive suggestions.
> We are encouraged that the Reviewer finds our problem interesting and practical, and our paper is well-written and well-demonstrated.
> We appreciate your acknowledgment of the contribution of new benchmark datasets and repair methods to the community.
>
> **Question:** *Do you think a much larger model will have the same limitation? Do you plan to experiment with larger models?, like the 16B parameter CodeGen*
>
> Thanks. To better answer your question, we have conducted extra experiments with CodeGen-16B-mono.
> We notice that, due to the limited time and computation, we provide the results with a subset of the original dataset instead. We will continue our experiment and incorporate results on all datasets in the final version.
>
> |buggy-HumanEval (200 instances) | pass@1| pass@10|pass@100|
> |:---:|:---:|:---:|:---:|
> |Clean prefix| 78.5 | 93.1 | 95.7 |
> |Buggy prefix| 8.9 | 20.4 | 25.2|
> |Removal| 43.2 | 70.9 | 80.1 |
> |Completion->rewriting| 14.4 | 29.7 | 35.6|
> |Rewriting->completion|70.9 | 80.2 | 86.9 |
>
>
> |buggy-FixEval (100 instances)| pass@1| pass@10|pass@100|
> |:---:|:---:|:---:|:---:|
> |Clean prefix| 59.7 | 72.6 | 75.7 |
> |Buggy prefix| 8.5 | 18.8 | 37.7 |
> |Removal| 20 | 49.8 | 68 |
> |Completion->rewriting| 8.6 | 18.4 | 37.7|
> |Rewriting->completion|8.9 | 19 | 32.9 |
>
> As we can see, CodeGen-16B-mono exhibits a similar phenomenon to our observations in the manuscript (with smaller versions), i.e., potential bugs make LLM harder to generate correct programs, and post-hoc techniques still suffer from a large gap to clean-prefix completion.
>
> **Question:** *How do the practice (rewriting user code is generally not expected) and the unavoidability of potential bugs affect the scope of the problem?*
>
> Great point! We do not commonly expect IDE smart tools to rewrite or change user code when recommending completions because no one has implemented such.
> We believe the work-in-progress code is draft, less refined, and more error-prone. Thus, it should be viewed as a hint of user intent rather than a golden one. From this view, it follows naturally if a pair programmer or a smart tool identifies or believes a certain part of the partial code is not what the developer meant, they should suggest the developer change their draft code rather than blindly continue it. From a user experience perspective, the IDE can display code change suggestions to a user’s existing code if the smart tool believes there is an error and some part of the code should be changed to something else. Similar functionality already exists for other types of code change suggestions, such as spelling correction suggestions or missing import suggestions.
>
> **Question:**  *More generally, how do you expect code completion applications to deal with buggy code?*
>
> Thanks for a great question.
> The response to the previous points described a potential user experience with smart completion tools. On the modeling side, we would like to see code completion solutions take more holistic views of the program specification, user intent, code and project context, and similar code and coding conventions into consideration rather than the narrow task of predicting what follows next from a string of code. This work is motivated by treating partial code not just as code context but as a hint of user intent. We explore the impact of “anti-patterns” induced by errors or drafts and present results on how we can possibly do better if we allow completion solutions to deviate from given code prefixes.
>
> —
> **Final notes**: We would like to thank the Reviewer for your constructive comments and appreciation of our findings. Per your suggestion, we will incorporate new experiments on a very large model and our answer about the scope of the problem and its impact on code completion applications. We hope our response makes you consider increasing your score and further support accepting our paper.

---

> > ### Comment · Reviewer_h8sG · 2023-08-17
> > **Thanks for the update. Increasing the score to Accept.**
> >
> > Thanks for the explanation.
> >
> > I am satisfied by the new experiment that you have added and your explanations.
> >
> > Because you have agreed incorporate the new experiment and discussion on problem scope, I am increasing the score to Accept.
> >
> > Good luck!

---

> > > ### Author Response · Authors · 2023-08-19
> > > **Re: Thanks for the update. Increasing the score to Accept.**
> > >
> > > Thank you for the positive feedback!

---

### Official Review · Reviewer_XA1K · 2023-07-07

**Soundness:** 3 good
**Presentation:** 3 good
**Contribution:** 3 good
**Rating:** 6
**Confidence:** 4

**Summary:**

Understanding the buggy code completion abilities of Large language models (LLMs) is a vital topic considering the increasing trend of adopting LLMs for programming assistance purposes. This paper studies the buggy code completion problem on different LLMs and datasets and takes a step to investigate three post-hoc methods to mitigate the effect of potential bugs. The authors evaluate the capabilities of four LLMs against buggy code with semantics changes and further compare the effectiveness of three repairing methods from multiple aspects.

**Strengths:**

+ Timely and vital problem.
+ The overall presentation of the paper is sound.
+ The experiments are conducted in a good manner.


### Significance:
This topic itself is motivating as the presence of potential bugs in code completion is common and unavoidable in program development. The authors construct two datasets for buggy code completion, which can be an asset for following research in this domain.


### Novelty:
Understanding the capabilities of LLMs on buggy code completion and corresponding repairing methods is novel.


### Verifiability and Transparency:
Although the replication is not yet available, the authors provide rich information in supplement material(appendix) to support the findings and discussions in the manuscript.


### Presentation:
Overall, the manuscript is written in a good manner, and I enjoyed reading this paper.

**Weaknesses:**

- Lack of justifications for some technical details.
- Some experiment results are not sufficiently explained.

This manuscript is generally easy to follow; however, some technical detail and experiment results still lack some explanations and justifications. Please see the comments below:

1. The definition of "potential bugs" is kind of ambiguous. In line 43, the authors state that "potential bugs are defined in relation to specific code completions and are not truly bugs per se without completions." However, during the dataset construction, the authors make candidate buggy samples by manipulating the semantic opposites, which can raise at least one violation regarding the test cases. These statements can somehow cause confusion about the definition of potential bugs.

2. In addition to the previous comment, the authors mention the injected bugs are not necessarily "incorrect." More justifications are needed to clearly explain the definition of the correctness of the bugs introduced in buggy code completion.

3. For the Buggy-FixEval dataset, the authors leverage a limit on the character-level edit distance between the clean prefix and the buggy one. Does that mean the constructed buggy prefix can have semantic changes at multiple spots? If so, this may raise more challenges for the following Rewriting->Completion repairing method since it contains a buggy line localization process.

4. The Rewriting->Completion needs more details and justifications specifically for the likelihood-based measurement used in buggy line selection. Although Appendix B.1 provides additional information for the likelihood measurement, no evaluation or discussion is found to justify the effectiveness of such measurement for line selection. Although the authors state a  heuristic oracle is used for comparison, limited information is available to give a concrete understanding. To the best of my knowledge, the token-level probability distribution in LLMs comes with relatively high predictive uncertainties, which may not be capable as an indicator for buggy line localization.

5. In the manuscript, the authors report the performance of different repairing methods with the metric of pass@1. From Appendix D Figure 6, I observe some interesting trends at pass@100 different from the performance on pass@1. Specifically, the Rewriting->Completion approach shows equivalently great or even beyond performance compared to the other two methods. The authors only discussed the findings for pass@1 in the manuscript; more insights from Figure 6 can be an asset to enrich the soundness of the paper.

6. In terms of the effect from bug and split locations, more in-depth discussion can help to deliver a better understanding of the characteristics of different repair methods. Namely, the authors may present some thoughts on why the Rewriting->Completion and Completion->Rewriting have distinct performances w.r.t different bug and split locations.


**Questions:**

1. Are there any experiments conducted to evaluate the effectiveness of the likelihood-based measurement on buggy line localization? Is this method effective on both single-line detection and multi-line section?


2. Does the constructed buggy prefix in Buggy-FixEval dataset can have semantic changes at multiple spots?


3. From Figure 6, are there any thoughts about the Rewriting->Completion method having a relatively better performance at higher k values?

**Limitations:**

The overall manuscript is sound. More implementation details about the dataset construction (Buggy-FixEva) and repair methods (Likelihood-based measures) can help to enhance the adaptability of the paper.

---

> ### Author Rebuttal · Authors · 2023-08-10
>
> We thank the Reviewer XA1K for the detailed review and constructive suggestions. We are very excited that you enjoyed reading our work and found our problem timely, vital, and motivating, as well as our manuscript well-written.
> We also appreciate your acknowledgment of the novelty and contribution of new benchmark datasets and repair methods to the community.
> Please find our detailed responses as follows.
>
>
> **Comment:** *... These statements (line 43 and data construction) can somehow cause confusion about the definition of potential bugs."*
>
> Thanks for the comment.
> During the construction of buggy-HumanEval, we alter the code prefix so that the prefix becomes buggy, aka fails test cases, with respect to the code suffix, aka the reference completion. The altered prefix is not itself “buggy per se” because bugginess is mostly pointless without a complete program. It IS buggy, however, “in relation to specific code completions” – in this case, the reference completion or the original suffix.
> We are revising these statements in the updated version.
>
>
> **Comment:** *More justifications are needed to clearly explain the definition of the correctness of the bugs introduced in buggy code completion.*
>
> Thanks for a good question. This again relates to the point that it’s pointless to say if the partial code is “buggy per se” or “incorrect by itself”. It can be incorrect with one completion but at the same time correct with a different completion. This ambiguity is inherent and is what exactly makes the problem difficult and interesting.
> We will add more justifications and examples to make the explanation more precise.
>
> **Question:** *Does the constructed buggy prefix in Buggy-FixEval dataset can have semantic changes at multiple spots?*
>
> Yes, theoretically speaking, the buggy prefix can have more than one semantic change. However, through our manually inspect, most buggy prefix contains only one potential bug, given the programs are mainly short.
> Regarding the rewriting->completion, we heuristically set a priority score for the line that appears early in the program among lines with high probabilities of containing bugs.
>
> **Question:** *Justifications for the likelihood-based measurement used in buggy line selection*
>
> * **Experimental results**: Thanks for the suggestion. The accuracies of localizing the line of potential bugs (with the same setting described in the manuscript) are approximately 82% and 53% on buggy-HumanEval and buggy-FixEval, respectively.
> For buggy-FixEval, we compare the detected line with the line of the first semantic difference between buggy and clean prefixes.
>
> * **comment 1**: “ … the token-level probability distribution in LLMs comes with relatively high predictive uncertainties, which may not be capable of as an indicator for buggy line localization.”
>
> > This is a good point. Indeed, our algorithm mitigates this variance by (i) instead of the softmax score, we measure the maximal margin between scores of the target token and the argmax token (similar to the popular approaches used in uncertainty quantification), and (ii) aggregate the score gap along the line and set a high threshold (to be more conservative).
> Our underlying idea is to treat the potential bug as the outlier in the generation flow of the LLM, and from our observation that most clean code has lower perplexity scores than the corresponding buggy code,  94% and 70% for HumanEval and FixEval, respectively.
>
> * **comment 2**: “ … Limited information is available to give a concrete understanding of heuristic oracle … “
>
> > Heuristic oracle only serves as a reference to practically compare our bug localization method.
> The oracle assumes access to the clean prefix to localize the bug line better. For instance, on buggy-HumanEval, it detects the potential bug by string-based comparing the buggy and clean prefixes to find the first difference.
> We will provide more details in our revision.
>
> **Question:** *3. From Figure 6, are there any thoughts about the Rewriting->Completion method having a relatively better performance at higher k values?*
>
> Thanks for a very good catch and suggestion.
> We hypothesize that higher k results reflect more about the "diversity" of the generated completions by methods. In particular,
> * *Removal* has no guidance about the completion, so the model tends to generate more diverse solutions. Thus, pass@1 can be high, but pass@k may be limited.
> * In *completion->rewriting*, only one prefix is used, making it harder for the repairer to fix if the buggy prefix steers LLM in the wrong direction.
> * *Rewriting->completion* balances the two: the rewriting phase first can help fix the prefix while still providing some good guidance to generate solutions that are more precise but less diverse (than removal), thus achieving better pass@k with higher k.
>
> We will incorporate these discussions in our revised version.
>
>
> **Comment:** *More discussion on bug and split locations. Why the Rewriting->Completion and Completion->Rewriting have distinct performances w.r.t different bug and split locations?*
>
> In Figure 3d, rewriting->completion achieves higher scores when potential bugs appear later. In contrast, completion -> rewriting (Figure 3c) achieves high scores when the code preﬁx is longer, indicating that this method can ﬁx potential bugs that appear early in the program. We suspect that this is because longer prefixes restrict more flexibility of the completion, making the generated completion less likely to deviate from the “mode” completion – the completion as if the injected bug was not there in the prefix. In this case, the completed code, aka the input to the subsequent code repairer, better resembles the kind of input the repairer is trained on and thus makes the repairer more likely to repair the program successfully.
>
> ---
> **Final notes**:  We are excited that you find our paper sound, and thanks again for carefully reading our work and for providing encouraging feedback!

---

> > ### Author Response · Authors · 2023-08-19
> > **Follow with the response**
> >
> > Dear Reviewer XA1K,
> >
> > We appreciate your time and effort in reviewing our paper. If any concern/question remains unaddressed, we are happy to provide further clarification.
> >
> > Thanks.

---

### Official Review · Reviewer_2xun · 2023-07-07

**Soundness:** 3 good
**Presentation:** 3 good
**Contribution:** 3 good
**Rating:** 5
**Confidence:** 3

**Summary:**

This paper introduces a new code-completion problem: potential bugs exist in the completion prefixes.  The authors constructed two benchmark datasets, one is created by manually injecting bugs from the HumanEval dataset, and the other is based on actual bugs in the FixEval dataset. Three baseline methods are proposed, including
(1) removing the prefix and then generating from scratch,
(2) completing at first and then repairing the whole solution, and
(3) repairing the prefix first and then finishing the completion.
The performances of different approaches on the two benchmark datasets are analyzed.

**Strengths:**

(1) The author proposes a new, challenging, and practical scenario in the field of code completion - how to complete code completion tasks in scenarios with potential errors in the context.

(2) Based on the setting of the paper, the author constructed two datasets based on HumanEval and FixEval as benchmark datasets.

(3) Combined with the code repair model, this paper proposes three baseline methods to deal with the problems in this scenario.

(4) The author conducted a comprehensive analysis of the performance of the baseline method.


**Weaknesses:**

(1) The baseline method provided in the paper is somehow straightforward, mainly a straightforward combination of code completion and repairing.

(2) The repairing process also introduces performance degradation, where a mechanism to judge how to use the code repair may be needed during the code completion.

(3) Although the authors argue that their new scenario is not a simple combination of code repair and completion, their baseline method is still a simple combination of the two tasks.

(4) The authors did not show the repairing performance among the two benchmark datasets.


**Questions:**

In Section 3.1.2, when constructing Buggy-FixEval dataset from FixEval dataset, the author split rejected solution in halves and regard their first halves as prefixes containing bugs, thereby creating samples with buggy prefixes. However, How to ensure the first half of the rejected solutions contain at least one bug? Please give more details.

**Limitations:**

(1) When constructing a dataset whose prefixes have potential bugs, the authors select the first half of rejected solutions in the FixEval dataset as prefixes with bugs. However, this process cannot ensure that the prefixes contain bugs. For example, bugs may exist in the second half of the solution.

(2) I am unsure if applying such three baseline methods to code completion will hurt the clean prefixes. The authors did not try to interpret this problem.

---

> ### Author Rebuttal · Authors · 2023-08-10
>
> We thank Reviewer 2xun for the detailed review and constructive suggestions. We appreciate your acknowledgment of the novelty, challenge, and practical setting of our problem and the contribution of new benchmark datasets, baseline methods, and our comprehensive analysis.
> Please find our answers to your comments and questions as follows.
>
> **Comment:** *proposed baselines are somehow simple, mainly a straightforward combination of code completion and repairing*
>
> Thanks for the comment. In fact, "simplicity" and "modularity" (a combination of existing modules) are our key designs for these baselines for understanding the current limits of code language models.
> These designs have some advantages: (i) easy to incorporate any LLM with the latest repairing modules without extra training, (ii) flexible to update each model separately, thus enabling more independent tests of the target LLM.
> Nevertheless, the rewriting->completion baseline is more sophisticated with an additional algorithm of bug localization.
>
> Furthermore, we are experimenting with two advanced techniques: (i) fine-tuned a CodeT5 pre-trained for neural program repair, and (2) few-shot in-context learning for fixing the program with loop repairing.
> We hope to release these investigations soon in a further study.
>
>
> **Limitation/Comment:** (i) *unsure if baseline methods hurt the clean prefixes*,  (ii) *"... may need a mechanism to judge how to use the code repair …"*
>
> This is a good point. Indeed, we acknowledged this phenomenon (baseline methods may hurt the clean prefixes) in Section 4.4. (see Tables 2 and 3). Furthermore, we showed our initial effort of controlling when to apply the repairing method by thresholding. For instance, in Table 3, given prior knowledge about the distribution of buggy programs, we may select a proper threshold to balance the clean and buggy settings.
>
> Moreover, we want to emphasize that the primary purpose of these baselines is to study the limits of code language models in our setting rather than **a perfect code completion solution**. Thus, we hope these are considered initial efforts for more advanced study of repairing techniques in future research.
> We will add more interpretations in the revised version.
>
>
>
> **Comment:** *No repairing performance among the two benchmark datasets*
>
> To our best interpretation of your comment, 'repairing performance’ means “repairing” without completion.
> If so, for *completion -> repairing* and *removal* methods, the repairing performance can be only measured by the final performance (pass@k) compared to the buggy-prefix completion, as shown in the manuscript.
> For the *rewriting -> completion* method, we can provide the accuracy of the line of bug localization as 82% and 53% on buggy-HumanEval and buggy-FixEval, respectively.
> Please let us know if this answer addresses your comment.
>
>
> **Question:** *In FixEval dataset, how to ensure the first half of rejected solutions contain bugs?*
>
> Thanks. This is a good question that sparked our discussion when designing the buggy-FixEval dataset.
> Indeed, we manually inspected all pairs to select the final 292 pairs that satisfy our conditions.
> Note that each pair has already been chosen to be similar (our supplementary A.2, page 2). We further execute the program that concatenates the buggy half with the completion half of the correct code to ensure that it fails at least 1 test case.
>
> ---
> **Final notes:** We are grateful that you find our idea novel and challenging. We will integrate our clarification into the final version. We hope our response helps you better understand our paper and consider increasing your score and further support accepting our paper.

---

> > ### Comment · Area_Chair_n3xV · 2023-08-18
> >
> > Hi Authors,
> >
> > Thank you for your response and we will take those into consideration.
> >
> > Best,

---

> > > ### Author Response · Authors · 2023-08-19
> > > **Follow with the response**
> > >
> > > Dear Reviewer 2xun and Area Chair,
> > >
> > > We thank the Area Chair n3xV for being so considerate.
> > >
> > > We appreciate your time and effort in reviewing our paper. We are happy to provide further clarification if there is any concern/question remaining unaddressed.
> > >
> > > Thanks.

---

### Author Rebuttal · Authors · 2023-08-10

We are thrilled that the reviewers appreciate our paper, and we want to extend our gratitude for their insightful feedback and constructive comments. These invaluable suggestions will undoubtedly enhance the quality of our work. First and foremost, it's encouraging to note that the reviewers have identified the following:
* (i) Our proposed problem is important, interesting, and practical (R-2xun, R-XA1K, R-h8sG, R-8UBb) with a novel setting (R-2xun, R-XA1K).
* (ii) Our evaluation is thorough, comprehensive, and well-conducted (R-XA1K, R-h8sG, R-Cn5Q) with rich insights and findings (R-2xun, R-XA1K, R-Cn5Q, R-8UBb).
* (iii) Our new public benchmark datasets are helpful for future research (R-2xun, R-h8sG, R-Cn5Q, R-8UBb).
* (iv) The mitigation techniques are intuitive and helpful in understanding the current limits (R-2xun, R-h8sG, R-8UBb).
* (v) Our paper is well-written and easy-to-follow (R-XA1K, R-h8sG, R-Cn5Q) with useful examples for illustration (R-Cn5Q).

We believe we have clarified and addressed all questions with our best. Please find our answers in each reviewer's section.

---

### Decision · Program_Chairs · 2023-09-21

**Decision:**

Accept (poster)

**Comment:**

The paper received 5/6/7/4/7 ratings. Reviewers acknowledged the paper is dealing with a timely code completion problem existed in the field. While two reviewers were concerned about the definition of the problem, those concerns were partially resolved during rebuttal discussion. One Reviewer found the experimental evaluation did not follow natural code bug distribution (imbalanced data). The AC agreed the problem exists, but it was also a common practice to show numbers on balanced data to avoid performance numbers being shadowed by the dominantly distributed category. After reading the paper, the reviews and rebuttal discussion. The AC stands with the majority votes and believes the the community will benefit from the paper.